# *Plasmodium falciparum* translational machinery condones polyadenosine repeats

**Slavica Pavlovic Djuranovic[1†]\*, Jessey Erath[1†], Ryan J Andrews[2],
Peter O Bayguinov[3], Joyce J Chung[4], Douglas L Chalker[4],
James AJ Fitzpatrick[1,3,5,6], Walter N Moss[2], Pawel Szczesny[7]\*, Sergej Djuranovic[1]\***

[1]Department of Cell Biology and Physiology, Washington University School of
Medicine, St. Louis, United States; [2]Roy J. Carver Department of Biochemistry,
Biophysics, and Molecular Biology, Iowa State University, Ames, United States;
[3]Washington University Center for Cellular Imaging, Washington University School
of Medicine, St. Louis, United States; [4]Department of Biology, Washington
University, St Louis, United States; [5]Department of Neuroscience, Washington
University School of Medicine, St. Louis, United States; [6]Department of Biomedical
Engineering, Washington University, St Louis, United States; [7]Institute of
Biochemistry and Biophysics Polish Academy of Sciences, Department of
Bioinformatics, Warsaw, Poland

**\*For correspondence:**
spavlov@wustl.edu (SPD);
szczesny.pawel@gmail.com (PS);
sergej.djuranovic@wustl.edu (SD)

[†]These authors contributed
equally to this work

**Competing interests:** The
authors declare that no
competing interests exist.

**Reviewing editor:** Nahum
Sonenberg, McGill University,
Canada

**Abstract** *Plasmodium falciparum* is a causative agent of human malaria. Sixty percent of mRNAs
from its extremely AT-rich (81%) genome harbor long polyadenosine (polyA) runs within their ORFs,
distinguishing the parasite from its hosts and other sequenced organisms. Recent studies indicate
polyA runs cause ribosome stalling and frameshifting, triggering mRNA surveillance pathways and
attenuating protein synthesis. Here, we show that *P. falciparum* is an exception to this rule. We
demonstrate that both endogenous genes and reporter sequences containing long polyA runs are
efficiently and accurately translated in *P. falciparum* cells. We show that polyA runs do not elicit any
response from No Go Decay (NGD) or result in the production of frameshifted proteins. This is in
stark contrast to what we observe in human cells or *T. thermophila*, an organism with similar AT-
content. Finally, using stalling reporters we show that *Plasmodium* cells evolved not to have a fully
functional NGD pathway.

## Introduction

The complex life cycle of *Plasmodium falciparum*, responsible for 90% of all malaria-associated
deaths, involves multiple stages in both the human and mosquito hosts. Asexual replication during
the intraerythrocytic development cycle (IDC) is tightly regulated over a 48 hr period, involving the
expression of the majority of its genes (*Gerald et al., 2011*; *Lu et al., 2017*; *Caro et al., 2014*). Pro-
gression through asexual stages (ring, trophozoite, schizont) of the IDC requires a strictly controlled
panel of gene expression profiles for each stage. A range of 16–32 daughter cells results from the
IDC. Thus, a single, originating merozoite must undergo several rounds of DNA synthesis, mitosis,
and nuclear division in a relatively short period (*Gerald et al., 2011*; *Lu et al., 2017*; *Caro et al.,
2014*). The apparent necessity for rapid and accurate DNA replication needs to be accompanied by
large-scale RNA transcription and protein synthesis. These processes occur during the 24 hr of tro-
phozoite stage before erythrocytic schizogeny and cytokinesis. Recent saturation-level mutagenesis
of the *P. falciparum* genome further emphasized and demonstrated that genes associated with cell

cycle control, translation, RNA metabolism, protein folding, and drug resistance are more likely to be essential for parasites fitness and survival (*Zhang, 2018*). However, a faithful execution of these fundamental processes is challenged by the extremely AT-rich *P. falciparum* genome: averaging ~81% in overall AT content. With a relatively small difference in AT-richness between the non-coding and coding regions; *P. falciparum* represents a unique case from other AT-rich organisms (*Glöckner, 2000*; *Szafranski et al., 2005*; *Zilversmit et al., 2010*; *Erath et al., 2019*). While the underlying reasons for such disproportionate representation of the four nucleotides in any given genome may be different, it is of vital importance that shifts towards extreme AT- or GC-richness must be accommodated by adaptation of the transcription and translation apparatuses; enabling the cell to transcribe and translate each gene appropriately.

Recently, it was demonstrated that the translation of genes with polyadenosine runs (polyA tracts), primarily coding for lysine residues, is attenuated in the majority of tested organisms presumably due to ribosomal stalling and frameshifting on such RNA motifs by action of ribosome-quality control complex (RQC) and mRNA surveillance mechanisms (*Ito-Harashima et al., 2007*; *Arthur et al., 2015*; *Arthur et al., 2017*; *Arthur and Djuranovic, 2018*; *Koutmou et al., 2015*; *Garzia et al., 2017*; *Juszkiewicz and Hegde, 2017*; *Sundaramoorthy et al., 2017*; *Tournu et al., 2019*; *Szádeczky-Kardoss et al., 2018*; *Chandrasekaran et al., 2019*; *Tuck et al., 2020*; *Tesina et al., 2020*). In human tissue cultures, the presence of just 12 adenosines in an mRNA-coding region was found to reduce the yield of protein synthesis by more than 40%, and runs of 30–60 adenosine nucleotides reduce protein synthesis by more than 90% (*Arthur et al., 2015*; *Arthur et al., 2017*; *Sundaramoorthy et al., 2017*). This effect on translation was observed in all tested organisms that include fruit flies (*Drosophila melanogaster*), yeasts (*Saccharomyces cerevisiae* and *Candida albicans*), plants (*Arabidopsis thaliana,* and *Nicotiana benthamiana*) and *E. coli* (*Zilversmit et al., 2010*; *Arthur et al., 2015*; *Koutmou et al., 2015*; *Tournu et al., 2019*; *Szádeczky-Kardoss et al., 2018*; *Tuck et al., 2020*); arguing for a universal response to coding polyA repeats. The consequence of translational arrest or slippage on polyA runs is the activation of RQC and one or more mRNA surveillance mechanisms, mainly No-Go (NGD) and Non-sense Mediated Decay (NMD) (*Arthur and Djuranovic, 2018*). High AU-content within transcript coding regions and an extreme AAA and AAU codon bias increase the propensity for polyA tracts in the *P. falciparum* transcriptome (*Saul and Battistutta, 1988*). Additionally, a 'just-in-time' transcriptional and translational model of gene expression during the relatively short trophozoite stage of the *P. falciparum* IDC (*Lu et al., 2017*; *Coulson et al., 2004*) make a rapid protein synthesis in an AU-rich transcriptome an appealing problem. While both, DNA and RNA polymerases must contend with high DNA AT-content, it is also puzzling what adaptations *P. falciparum* have made to its translational machinery to overcome the unusual AU-richness of mRNAs, which would affect the fidelity and efficacy of protein synthesis. With 'just-in-time' translation of numerous A-rich coding sequences (*Szafranski et al., 2005*; *Erath et al., 2019*) and poly-lysine proteins harboring an AAA codon bias, expressed at all stages in the parasite life cycle (*Lu et al., 2017*; *Caro et al., 2014*; *Bunnik et al., 2013*), the *Plasmodium* translation machinery would represent an exception in protein synthesis compared to other organisms.

Here we present data that indicate that the *P. falciparum* translational machinery and its NGD pathway have adapted to translate long runs of polyadenosine nucleotides into poly-lysine repeats. Using comparative bionformatic analyses, we show that malaria parasites contain an unusually high numbers of polyA-tract-containing genes compared to other eukaryotic organisms. We find this to be a common feature of all analyzed *Plasmodium* species regardless of their AT-content, arguing for evolutionary conservation of such sequences in *Plasmodium* genomes. Expression of endogenous genes and reporters with polyA tract motifs in *P. falciparum* cells results in efficient and accurate protein synthesis in direct contrast to what we observe for human-tissue cultures and *T. thermophila* cells. We find no evidence for either ribosomal stalling or frameshifting during the translation of long polyadenosine runs in *P. falciparum* cells. Interestingly, induction of the NGD pathway by either reporter with stable RNA structure or isoleucine starvation results in reduced reporter protein levels but without any detectable changes in mRNA levels, arguing for alterations in the NGD pathway. Finally, our analysis of *P. falciparum* ribosome structure suggests a model whereby multiple changes may have evolved to accommodate the unusual AU-richness and high percentage of poly–lysine runs of the *P. falciparum* transcriptome and proteome, respectively.

## Results

### *Plasmodium species:* A paradigm-breaking genus

Previous studies indicated that polyA runs in the coding sequences serve as hurdles to translation (*Ito-Harashima et al., 2007*; *Arthur et al., 2015*; *Arthur et al., 2017*; *Arthur and Djuranovic, 2018*; *Koutmou et al., 2015*; *Garzia et al., 2017*; *Juszkiewicz and Hegde, 2017*; *Sundaramoorthy et al., 2017*; *Tournu et al., 2019*; *Szádeczky-Kardoss et al., 2018*) that efficiently reduce protein yield and initiate NGD to degrade the mRNA. Given the very high AT-content in certain eukaryotic species (*Glöckner, 2000*) we sought to explore the association between coding region AT-content and transcript polyA-tract-motif abundance. In doing so, we analyzed 152 eukaryotic genomes (*Figure 1A*). We focused on stretches of polyadenosine nucleotides, or as defined previously 12A-1 motifs; sequences with a minimal length of 12 A's allowing for one mismatched base (*Arthur et al., 2015*). We settled on 12A-1 sequence pattern since the presence of this motif reduces protein production by 40–60% in multiple human genes (*Arthur et al., 2015*; *Arthur et al., 2017*; *Tuck et al., 2020*). Subsequent analyses indicated that the reduction in protein amounts for genes with 12A-1 motif could be attributed to ribosome stalling and frameshifting (*Arthur et al., 2017*; *Koutmou et al., 2015*). Analyses of the selected set of eukaryotic genomes indicate that *P. falciparum* and other members of the *Plasmodium* genus have a much higher ratio of polyA tract genes when normalized to genomic AT-content (*Figure 1A*).

Interestingly, this feature of *Plasmodium* species is conserved regardless of their genomic AT-content, resulting in two groups (low and high AT-content *Plasmodium spp.*) with unusually large portions of polyA tract genes ranging from 35 to over 65% of the total transcriptome (*Figure 1A*; *Erath et al., 2019*). These two groups, which also appear to be separated geographically, demonstrate that even with a reduction in genomic AT-content, selective pressure exists for *Plasmodium spp.* to maintain polyA tract motifs. However, regardless of a selective reduction of AT-content in some *Plasmodium* species, or an almost complete lack thereof in the case of avian *Plasmodium* counterparts (*Videvall, 2018*), the trend of harboring a high ratio of polyA-affected transcripts given a particular AT-richness, as well as their conservation within a significant number of genes across species, remains a paradigm-breaking hallmark of the genus.

To further emphasize the differences between *P. falciparum* and other organisms, we analyzed the number of genes that harbor polyA runs as well as the total length of consecutive adenosine nucleotides in transcripts (*Figure 1—figure supplement 1* and *Figure 1B*, respectively). *P. falciparum* showed a significantly higher amount of genes with polyA runs of ≥12 A's in a row when compared with its human and mosquito (*Anopheles gambiae*) hosts (*Figure 1—figure supplement 1*). Over 63% of genes in *P. falciparum* have polyA tract motifs, while only 0.7–2% of total genes in humans and mosquito genomes, respectively, contain these motifs. We used the most recently updated Ensembl database to look into coding sequences of lab strain of malaria *P. falciparum 3D7*, *H. sapiens,* and *T. thermophila* (*Hunt et al., 2018*). While there is already a significant difference in the number of genes with eleven consecutive adenosines (*P. falciparum 3D7* – 1555, *T. thermophila* - 364 and *H. sapiens* - 77 genes), the most striking sequence difference is associated with 16 consecutive adenosines (16As). The *P. falciparum 3D7* genome contains 188 genes with 16As, while the human host and AT-rich *T. thermophila* together have only three genes (*T. thermophila* two and *H. sapiens* 1) (*Figure 1B*). Additionally, *T. thermophila* a protozoan with an overall 73% AT-rich genome and 24,000 predicted genes, had almost nine times less genes with twelve consecutive adenosines than *P. falciparum*, 172 and 1526, respectively. *Ensembl* database reports 105 *P. falciparum 3D7* genes with exactly 18As and more than 150 genes with more than 20 consecutive adenosines (20-65As, *Figure 1B*). Analyses of *P. falciparum* genomes in the PlasmoDB database (*Beznosková et al., 2013*), curated explicitly for malaria parasite species, report more than double that number with 329 genes having runs of ≥20 consecutive adenosines. Most of these genes appear to be annotated as *Plasmodium* specific. However, PlasmoDB also shows genes with up to 59 consecutive adenosines in conserved pathways, such as the phosphatidylinositol-4-phosphate 5-kinase gene (PfML01_010014200), which is involved in the inositol metabolism and signaling pathways (*Beznosková et al., 2013*). Finally, the maximum length of consecutive adenosine runs in coding sequences of *Plasmodium* genomes indicated 111 As in *P. falciparum fch four* strain with the human host or 132 As in *P. reichenowi*, strain causing chimpanzee malaria (*Erath et al., 2019*; *Habich et al., 2016*). As such, the length of polyA runs in coding sequences of malaria parasites exceeds the

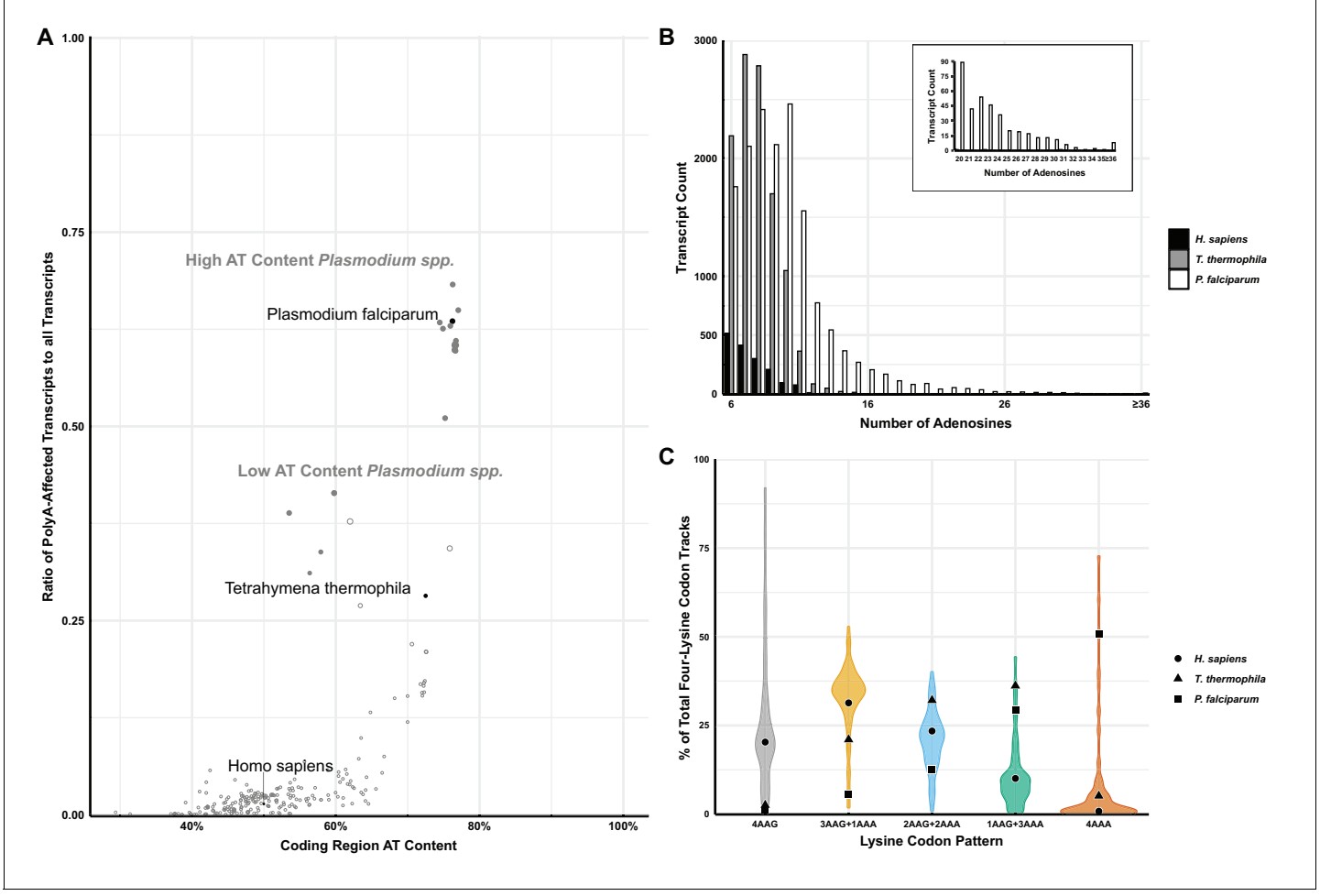

**Figure 1.** Analyses of polyA tracts in eukaryotic genomes. (**A**) The plot of 152 species representing a comparison of the ratio of polyA-affected transcripts (over a total number of transcripts) to the AT content of the coding region for each organism. *H. sapiens*, *T. thermophila*, and *P. falciparum*, as organisms pertinent to this paper are in black. For reference, other model organisms of interest are displayed in gray, including a position of high (65% average) and low (35%) AT-content *Plasmodium spp.* (**B**) Transcript counts for genes with 6 to 36 consecutive adenosines for *H. sapiens, T. thermophila,* and *P. falciparum. H. sapiens* and *T. thermophila* are limited to a single transcript at length of ≤17 As. The longest *P. falciparum* 3D7 transcript reaches maximal 65As, with multiple transcripts of ≤36 As. (**C**) Violin plot of lysine codon usage distribution in tracts of four lysine residues for 152 organisms. 3AAG+1AAA, 2AAG+2AAA and 1AAG+3AAA indicate different ratios of AAG and AAA codons in runs of four consecutive lysine codons. 4AAG and 4AAA indicate poly-lysine runs with only AAG or AAA codons, respectively. *H. sapiens* (circle), *T. thermophila* (triangle), and *P. falciparum* (square) are specifically noted.

The online version of this article includes the following source data and figure supplement(s) for figure 1:

**Source data 1.** Lysine codons distribution in 4xLys runs in eukaryotic genomes.

**Figure supplement 1.** Percentage of genes with ≥12A (white) and ≥12A-1 (gray) consecutive adenosine nucleotides for each organism.

length of the average size of 3'UTR polyA tails found on the majority of transcripts in eukaryotic species (*Subtelny et al., 2014*; *Brown and Sachs, 1998*; *Chang et al., 2014*). Translation of 3'UTR located polyA tail in eukaryotic organism activates yet another mRNA surveillance pathway - non-stop decay (NSD) (*Chandrasekaran et al., 2019*; *Tesina et al., 2020*; *Shoemaker and Green, 2012*).

The disparity in the number of genes with polyA tracts could be due to previously observed codon biases in *P. falciparum* (*Saul and Battistutta, 1988*). However, it was already shown that codon bias and tRNA abundance do not correlate with codon selection in genes coding for lysine repeats (*Arthur et al., 2015*; *Koutmou et al., 2015*). The poly-lysine repeats in proteins of the analyzed model organisms are usually encoded by AAG codons. To investigate the distribution of AAA and AAG codons in poly-lysine tracts in more detail, we analyzed transcripts from *P. falciparum* and

152 other eukaryotic genomes (*Figure 1C*). We observed a complete reversal of the trend exhibited in other organisms, including humans and the AT-rich *T. thermophila*. *P. falciparum* had the highest abundance of transcripts hosting four consecutive AAA codons in runs of four lysine residues (*Figure 1C*). This divergence from other analyzed transcriptomes is preserved in other members of *Plasmodium spp.*, with *P. berghei* being an extreme example using only AAA codons in 68% of transcripts coding for poly-lysine runs. Finally, we analyzed the biological function and essentiality of the polyA tract and poly-lysine containing genes in *P. falciparum* genome. Previous analysis indicated that 70–85% of orthologs of polyA tract carrying genes from *P. falciparum* have the same polyA motifs in genes from other *Plasmodium* species, regardless of their genomic AT content (*Habich et al., 2016*). The high degree of the conservation of polyA tracts has also been noted for other eukaryotic organisms (*Arthur et al., 2015*). A majority of *Plasmodium* polyA tracts genes and poly-lysine proteins fall into the group of essential genes based on the recent mutagenesis studies (*Zhang, 2018*). This outcome is expected given that gene ontology results indicate enrichment in gene products involved in crucial cellular processes such as protein synthesis, RNA biogenesis, and chromosome segregation. The same gene ontology groups were previously observed in poly-lysine repeats and polyA tract genes of the other organisms (*Arthur et al., 2015*). In addition to these groups, *Plasmodium* species also had enrichment of polyA tract motifs in a group of genes defined as the cellular and pathological cell adhesion ontology group (*Supplementary file 1*). As such, our overall bioinformatic analyses demonstrate that *Plasmodium* genomes represent a unique set of organisms that have an enrichment of polyA tracts in the coding sequences. The overall conservation of both polyA tracts in the transcriptome and poly-lysine repeats in the proteome of *P. falciparum* has been evolutionary selected and conserved due to possible benefits for the parasite.

## Reporters with polyA tracts are not a problem for *P. falciparum*

As mentioned earlier, polyA tracts and poly-lysine repeats cause a reduction in mRNA stability and protein amounts, respectively, due to the ribosomal stalling and frameshifting on such motifs (*Ito-Harashima et al., 2007*; *Arthur et al., 2015*; *Arthur et al., 2017*; *Arthur and Djuranovic, 2018*; *Koutmou et al., 2015*; *Garzia et al., 2017*; *Juszkiewicz and Hegde, 2017*; *Sundaramoorthy et al., 2017*; *Tournu et al., 2019*; *Szádeczky-Kardoss et al., 2018*; *Tesina et al., 2020*). To investigate further how the AU-rich *P. falciparum* transcriptome with multiple polyA tracts gets effectively translated, we used double HA-tagged reporter constructs. The 36 adenosine nucleotide (36 As) insertion, coding for 12 lysine residues, was inserted between the sequence of double HA-tag and a fluorescent protein (+polyA$_{36}$, *Figure 2—figure supplement 1*). As a control, we used a reporter that had only double HA-tag in front of the fluorescent-protein sequence (-polyA$_{36}$, *Figure 2—figure supplement 1*). We expressed our reporter constructs from plasmid vectors in the *P. falciparum* Dd2 lab strain. In parallel, we expressed the same constructs in human dermal fibroblasts (HDFs) and *T. thermophila* cells (*Figure 2A and B*). We followed mRNA abundance of each construct by qRT-PCR (*Figure 2A*), and expression of the reporter protein was followed by western blot detection of the double HA-tag in all three organisms (*Figure 2B*). We observed robust changes in normalized mRNA levels (*Figure 2A*) and substantial losses in protein expression (*Figure 2B*) for reporters with polyA tracts (+polyA$_{36}$) in both HDFs and AT-rich *T. thermophila* (*Arthur et al., 2017*; *Arthur and Djuranovic, 2018*). In comparison to HDFs and *T. thermophila*, we observed minimal, if any, effects from polyA tract insertion on reporter mRNA and protein expression in *P. falciparum* cells (*Figure 2A and B*). Further analysis of *P. falciparum* cells by live-fluorescence microscopy confirms the equivalent expression of mCherry reporter, judging by the intensity of fluorescence between constructs with and without polyA tract (+polyA$_{36}$ and –polyA$_{36}$, *Figure 2C*).

To assess whether the efficiency of protein synthesis is altered when the polyA tract is located further downstream of the start codon, we designed a construct with thioredoxin (Trx) and nano-luciferase (nanoluc) proteins separated with a double HA-tag and 36As coding for a poly-lysine run (*Figure 2—figure supplement 2*). Measurement of nanoluc luminescence from the same number of drug-selected parasites indicates similar expression of a reporter with a polyA tract (+polyA$_{36}$) compared with the control reporter (-polyA$_{36}$, *Figure 2D*). We observed the same ratio when we analyzed the expression of reporters using western blot analysis (*Figure 2D* and *Figure 2—figure supplement 2*), arguing that the position of polyA tracts in coding sequence does not influence the efficiency of protein synthesis in *P. falciparum* cells.

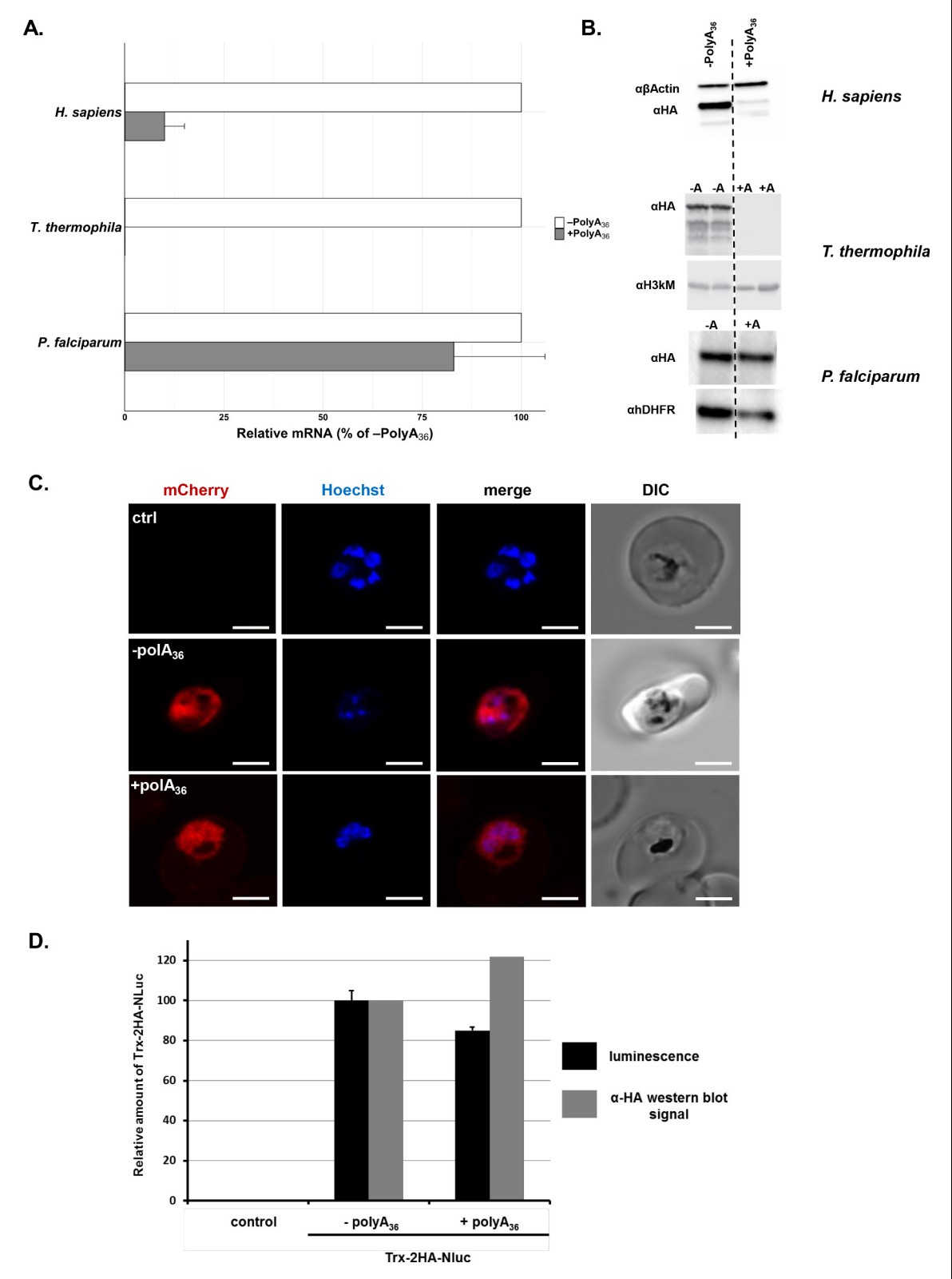

**Figure 2.** *P. falciparum* cells express reporters with long polyA tracts. (**A**) mRNA abundance of reporter constructs (+polyA36) by qRT-PCR relative to their counterpart lacking polyA stretches (-polyA36) in *H. sapiens, T. thermophila,* and *P. falciparum* cells. Data represent three biological replicates with a standard deviation. (**B**) Expression of reporter constructs in *H. sapiens, T. thermophila,* and *P. falciparum* followed by western blot analysis with αHA or αGFP antisera. Samples from two integrated clones for the -polyA36 control (-A) and the +polyA36 reporter (+A) are shown for *T. thermophila*. αβ-
*Figure 2 continued on next page*

*Figure 2 continued*

actin, α-Histone H3 trimethyl-lysine (H3kM) and αhDHFR are used as loading controls for western blot analysis from *H. sapiens*, *T. thermophila* and *P. falciparum* cells, respectively. (C) Images from live fluorescence microscopy of *P. falciparum* expression of reporter constructs with (+polyA$_{36}$) and without (-polyA$_{36}$) polyA tracts as well as parent (non-transfected) line, 2.5 μm scale bar. (D) Quantification of protein amounts for Thioredoxin-2HA-NanoLuciferase (Trx-2HA-NLuc) reporter without (-polyA$_{36}$) and with 36 adenosine stretch (+polyA$_{36}$) expressed in *P. falciparum cells*. Western blot analysis of Trx-2HA-NLuc reporter (*Figure 2—figure supplement 2*) and luminescence measurements were normalized to hDHFR or cell number, respectively. Luminescence data represent the mean value of three biological replicates with standard deviation.

The online version of this article includes the following source data and figure supplement(s) for figure 2:

**Source data 1.** Luminescence and western blot quantification data.
**Figure supplement 1.** Generalized scheme of reporter constructs used for expression in *H. sapiens*, *T. thermophila*, and *P. falciparum*.
**Figure supplement 2.** Generalized schematic of Thioredoxin fusion NanoLuc reporter construct used for episomal expression in *P. falciparum* cells (Trx-2HA-Nluc).
**Figure supplement 3.** HA-pull-down assay of –polyA36, +polyA36 reporters.

Since polyA tracts cause ribosomal frameshifting (*Arthur et al., 2015*; *Koutmou et al., 2015*; *Sundaramoorthy et al., 2017*) we analyzed the expression of our nanoluc and fluorescent reporters for the possible presence of frameshifted protein products. Western blot analyses of nanoluc reporters did not indicate the HA-tagged protein band at 14kD that would represent frameshifted nanoluc (*Figure 2—figure supplement 2*). Moreover, the immunoprecipitation of mCherry reporters showed only equivalent amounts of the full-length product (*Figure 2—figure supplement 3*). There is still a possibility that ribosomal frameshifting occurred in polyA tract reporters, but the protein product was unstable in *P. falciparum* cells, and we failed to detect it. However, if such frameshifting events did happen, they did not reduce overall levels of the full-length protein. Finally, to check whether polyA tract constructs resulted in the synthesis of poly-lysine peptides, we analyzed all constructs using 2D PAGE electrophoresis (*Figure 3*). This analysis indicated a shift in isoelectric point (pI) of approximately one pH unit for the (+polyA$_{36}$) construct compared to the wild-type one (-polyA$_{36}$), 7.2–7.4 and 6.25–6.5, respectively. This shift on the 2D PAGE gels was the same for construct with 12xAAG lysine codons and is expected based on the calculated isoelectric points if all 12xAAA or 12xAAG codons were translated into lysine residues. Taken together, our analyses of different reporter expression data (*Figures 2* and *3*, *Figure 2—figure supplements 2* and *3*) indicates that polyA tracts are tolerated by the parasite translational machinery, without apparent effects on either stability of mRNA or quality of synthesized protein.

## Endogenous polyA tract genes are efficiently expressed in *P. falciparum*

Due to the potential for negative selection against polyA tracts (*Guler et al., 2013*), particularly in laboratory conditions, we also wanted to investigate how *P. falciparum* translates endogenous genes with polyA tracts. With as much as 60% of the parasite transcriptome harboring polyA tract motifs, we performed a comparative analysis of ribosome profiling data from *P. falciparum* (*Caro et al., 2014*) and aggregated data for human tissues conveniently harmonized at GWIPS database (*Michel et al., 2014*). We analyzed whether endogenous polyA tracts and poly-lysine sequences induce translational pausing in both sets of data. Ribosome stalling can be observed in the ribosome profiling data as an increase in the abundance of ribosome footprints on sequences that cause ribosomes to pause during translation (*Ingolia et al., 2014*). Cumulative data for all transcripts with polyA tracts from human cells indicate substantial translational pausing on these sequences (*Figure 4A*). The same effect was noted on poly-lysine sequences that contained more than four consecutive lysine residues in multiple other studies using *S. cerevisiae* ribosome profiling datasets without cycloheximide treatment or datasets from human tissue cultures regardless of cycloheximide treatment (*Arthur et al., 2015*; *Guydosh and Green, 2017*; *Requião et al., 2016*; *Charneski and Hurst, 2013*). However, analyses of previously published *P. falciparum* ribosome profiling dataset (*Caro et al., 2014*) indicated no evidence for ribosome stalling in *P. falciparum* transcripts containing polyA tracts. Normalized ribosome occupancy for *P. falciparum* transcripts with a length of less than or equal to 22 consecutive adenosine nucleotides (≤22As), that code for more than seven consecutive lysines, indicated more or less equal ribosome occupancy over polyA tract (*Figure 4B*). We limited our ribosome profiling analyses of *P. falciparum* transcripts to polyA tracts of ≤22As, since the

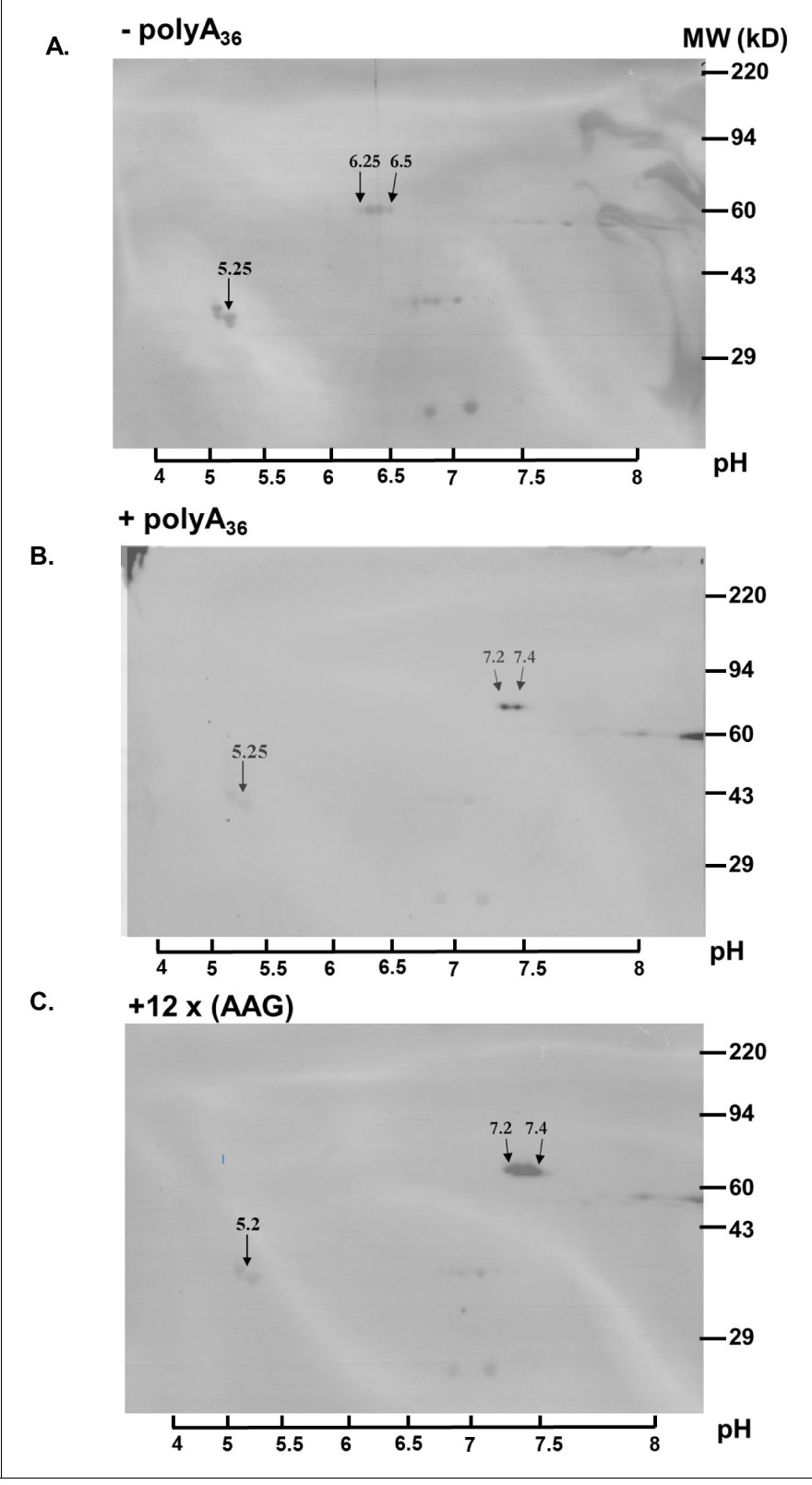

**Figure 3.** Poly-A tracts are correctly translated in *P. falciparum* cells. (**A**) 2D gel analysis of HA-IP samples of wild type reporter (-polyA$_{36}$). The western blots show isoelectric point (pI) at 6.25 and 6.5 (arrows). (**B**) 2D gel of reporter with polyA tract (+polyA$_{36}$) sample indicates pI 7.25 and 7.4 (arrows).( **C**) 2D gel analyses of reporter with twelve lysines coded by AAG codons (12 x AAG) indicates the same effect on pI value, pI is 7.25 and 7.4 (arrows). Overlay of images of PVDF membranes stained with Coomassie Brilliant Blue, and images of western blots probed with HA-antibody. The scale with pH

*Figure 3 continued on next page*

*Figure 3 continued*
is on the bottom. Coomassie-stained membranes show markers (right side) and tropomyosin (33 kDa) and pI at 5.2 (arrow) as an internal standard for Isoelectric focusing (IEF).

cumulative transcript analyses become hindered by the low sequence complexity of the region, or reduced number of reads, for the long polyA tracts (*Figure 4—figure supplement 1*). As such, the dip in *P. falciparum* occupancy plot is a result of a reads mapping artifacts. The longer polyA segment, the harder it is to uniquely map a read - typically such reads are discarded. This artifact is clearly seen in the *Figure 4—figure supplement 1*, where the occupancy is shown per length of polyA segment - the larger polyA track, the larger symmetrical gap around position 0. The correlation holds true to about 30nt, which is the read length in the original experiment by *Caro et al., 2014*.

Since there is a massive increase in protein synthesis in *P. falciparum* cells during the trophozoite and schizont stage of IDC (*Caro et al., 2014*; *Bunnik et al., 2013*) we analyzed whether there is a different distribution of ribosome profiling reads surrounding polyA tracts at different stages of *P. falciparum* development. We found that the translation of endogenous polyA tracts is independent of different stages of *P. falciparum* IDC, as we did not notice any significant difference in the distribution of ribosome protected fragments (*Figure 4—figure supplement 2*) and ostensibly irrespective of the length of polyA tracts or poly-lysine runs. While we observe a relatively small increase in the number of elongating ribosomes on polyA segments in the late trophozoite and schizont stages of IDC (*Figure 4—figure supplement 2*), it is unclear if these indicate higher protein expression or are just an artifact of a general increase in protein synthesis at this IDC stages.

In parallel with analyses of ribosome profiling data, we also selected a small subset of polyA tract-containing genes for independent expression analysis. We followed PfGAPDH (PF3D7_1462800) as a control gene without polyA tract, and *Pf*ZFP (PF3D7_1464200 (three polyA tracts and run of 20 consecutive adenosines), *Pf*CRK5 - PF3D7_0615500 (four polyA tracts and run of 20As) and PfIWS1L - PF3D7_1108000 (three polyA tracts and run of 31As), to represent genes with polyA tracts (*Figure 4—figure supplement 3*). *Pf*ZFP is a zinc-binding protein with homology to the human ZC3H6 gene (Zinc Finger CCCH Domain-Containing Protein 6), *Pf*CRK5 is cdc2-related protein kinase five and PfIWS1L is transcription elongation factor associated with RNA Polymerase II. We used a sorbitol based method to synchronize *P. falciparum* Dd2 strain at the ring stage of IDC and followed transcript profiles over the next 48 hr of the parasite life cycle. The mRNA profiles of three polyA tract genes were analyzed using qRT-PCR and normalized to the expression of *Pf*GAPDH in *P. falciparum* Dd2 lab strain. A time-course study of synchronized parasite culture indicated that the selected polyA tract transcripts are efficiently transcribed at all-time points when compared to the control gene (*Figure 4—figure supplement 4*). Moreover, the increase in transcription of all three polyA tract genes followed previously described just-in-time transcription profile in trophozoite and schizont IDC stage of *P. falciparum*.

Finally, mRNA translation could also be affected by the ability of RNA to fold into unique functional structures; for example RNA secondary structures have been shown to coordinate ribosomal frameshifting (*Kim et al., 2014*; *Mouzakis et al., 2013*), slow ribosomal progression to allow protein folding (*Faure et al., 2016*) and to affect ribosomal density (*Zur and Tuller, 2012*; *Andrews et al., 2017*). To determine the RNA structural characteristics of the *P. falciparum* transcriptome, we analyzed the ability of all coding sequences to form thermodynamically stable RNA secondary structures using an in silico approach to characterize RNA folding landscapes (*Andrews et al., 2017*; *Andrews et al., 2019*). We compared the relative structural stability of the *P. falciparum* transcriptome to that of humans and the AU-rich *T. thermophila*. On average, the coding sequences from *T. thermophila* and *P. falciparum* yielded structures with higher predicted minimum free energy (MFE; *Figure 4—figure supplement 5*), than those from the coding sequences of human genes; a result which mostly correlates with the higher overall AU-content of *P. falciparum* and *T. thermophila* transcripts. In this analysis, each MFE is also characterized by a thermodynamic z-score (see Materials and methods). The thermodynamic z-score normalizes for nucleotide content and suggests whether sequences may form potentially functional structures (*Clote et al., 2005*). Specifically, the thermodynamic z-score indicates whether a nucleotide sequence has been ordered to adopt a more stable secondary structure than its nucleotide content would typically produce (*Clote et al., 2005*)

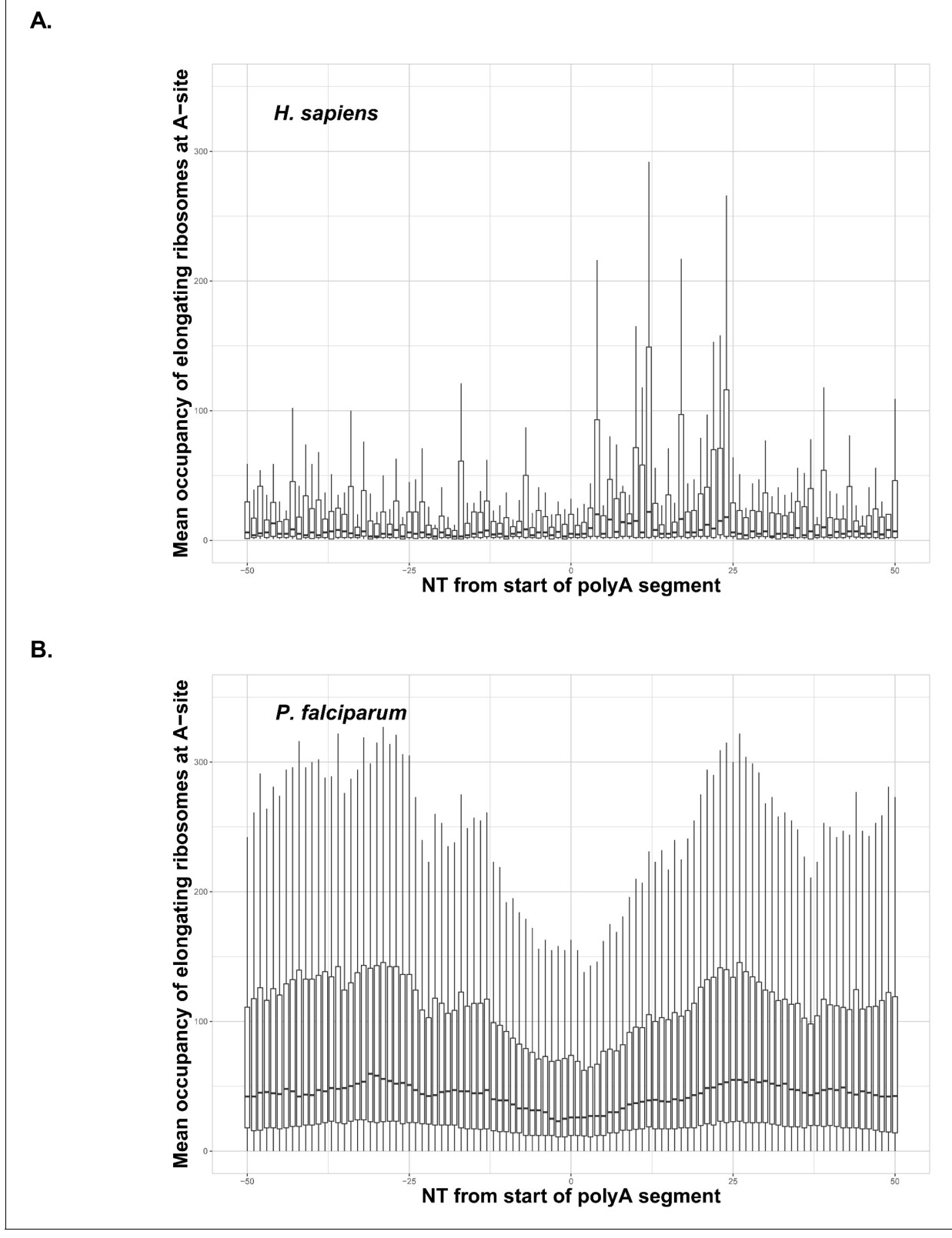

**Figure 4.** Occupancy of elongating ribosomes (mapped to A-site) around the start of polyA segments in human (**A**) and *P. falciparum* (**B**). Occupancy is shown on the same scale. In both cases, to avoid the inclusion of sparsely mapped segments, regions with average occupancy below the mean for the whole dataset were excluded. In the case of *P. falciparum* gene segments with polyA tracts shorter than 22 adenosine nucleotides were taken into account. Centerlines show the medians; box limits indicate the 25th and 75th percentiles; whiskers extend 1.5 times the interquartile range from the

*Figure 4 continued on next page*

*Figure 4 continued*

25th and 75th percentiles. Dip in around position 0 in the *P. falciparum* plot is an artifact of the reads mapping procedure shown in ***Figure 4—figure supplement 1***. It is not observed in human data, because of shorter polyA segments and lower occupancy overall around these segments.

The online version of this article includes the following figure supplement(s) for figure 4:

**Figure supplement 1.** Ribosome occupancy around polyA segment of *P. falciparum* transcripts (summarized across all life stages) and grouped by polyA segment length (12–68 adenosine nucleotides in a row transcripts are shown).

**Figure supplement 2.** Occupancy of elongating ribosomes (mapped to A-site) around start of polyA segment in *Plasmodium* at different life stages.

**Figure supplement 3.** Sequences of three *P. falciparum* polyA tract genes (PF3D7_1464200, PF3D7_0615500, PF3D7_1108000).

**Figure supplement 4.** Time course mRNA expression analysis of three genes (PF3D7_1464200, PF3D7_0615500, PF3D7_1108000) containing polyA stretches of varying lengths (largest: 20, 20, and 31 adenosines respectively) normalized to GAPDH (PF3D7_1462800) starting with highly synchronized rings at time zero.

**Figure supplement 5.** Box and whisker plots are showing the distribution of mean folding energy values (kcal/mol; measures the stability of RNA structure) calculated for each coding sequence from *H. sapiens*, *T. thermophila*, and *P. falciparum*, resulting from a scanning window analysis (see Materials and methods).

(which is the case for many ncRNAs (*Freyhult et al., 2005*) as well as RNA regulatory structures embedded in mRNAs *O'Leary et al., 2019*; *Andrews et al., 2018*).

Interestingly transcriptome z-scores revealed no significant differences between *P. falciparum* and *T. thermophila* (*Figure 4—figure supplement 5*). This result indicates that while RNA folding stability varies between species, following from skews in nucleotide content, the potential occurrence of ordered, structured motifs does not. This suggests that globally, *P. falciparum* coding sequences do not appear ordered to adopt structures any more stable than would be expected for random sequences or with the same nucleotide content found *in T. thermophila*. However, expression of reporters with polyA-tracts in *T. thermophila* is strongly attenuated (*Figure 2*). This computational effort does not take into account the most recent work describing the m6A mRNA methylation dynamics within *P. falciparum* coding sequences (*Baumgarten et al., 2019*). The m6A modification may impact RNA structure and association of *P. falciparum* mRNAs with certain RNA binding proteins (*Liu et al., 2015*). However, previous work has demonstrated that m6A methylation does not affect the ability of the ribosome to decode mRNAs but instead has a modest impact on the rate of translation (*Hudson and Zaher, 2015*; *Choi et al., 2016*), which could in-turn reduce the rate of protein synthesis in *P. falciparum*. Taken together, our analyses of the *P. falciparum* transcriptome and translation of genes with endogenous polyA tracts, as well as in silico assessment of overall structure and stability of AU-rich transcripts indicates that the *P. falciparum* translation machinery adapted to long and numerous polyA tracts in coding sequences of the majority of genes.

## NGD pathway is not connected to mRNA degradation in *P. falciparum*

PolyA tracts and poly-lysine repeats are highly efficient at causing ribosome stalling and frameshifting in bacteria and most eukaryotes (*Ito-Harashima et al., 2007*; *Arthur et al., 2015*; *Arthur et al., 2017*; *Arthur and Djuranovic, 2018*; *Koutmou et al., 2015*; *Garzia et al., 2017*; *Juszkiewicz and Hegde, 2017*; *Sundaramoorthy et al., 2017*; *Tournu et al., 2019*; *Szádeczky-Kardoss et al., 2018*; *Chandrasekaran et al., 2019*; *Tuck et al., 2020*; *Tesina et al., 2020*). However, our analyses of endogenous *P. falciparum* genes and reporters with long polyA runs, as well as immunoprecipitations did not indicate any changes in either mRNA stability, protein amounts, or protein quality (*Figures 2* and *3*) that was previously observed in *E. coli*, yeast, and human cells (*Arthur et al., 2015*; *Arthur et al., 2017*; *Koutmou et al., 2015*; *Tuck et al., 2020*). So, we turned towards analyses of mRNA surveillance pathways in *P. falciparum* cells and their contribution to the synthesis of poly-lysine peptides from long polyA tracts. The NMD pathway seems to be intact in *Plasmodium* cells (*Sorber et al., 2011*), while the existence of the NGD pathway has not been previously tested. We initially focused our analysis on two proteins that have been documented to be crucial for NGD pathways in eukaryotes, Hbs1 and Pelota (*Shoemaker and Green, 2012*). The Pelota protein in complex with Hbs1 recognizes and rescues stalled ribosomes with an empty A site on long polyA stretches, absence of tRNAs during starvation, damaged mRNAs, or stable RNA structures (*Shoemaker and Green, 2012*; *Guydosh and Green, 2017*; *Tsuboi et al., 2012*; *Hilal et al., 2016*; *Becker et al., 2011*). Our bioinformatics search of the *P. falciparum* genome did not identify an apparent homolog of Hbs1 nor the recently reported *S. cerevisiae* endonuclease Cue2 (NONU-1 in *C. elegans*)

(**D'Orazio et al., 2019**; **Glover, 2019**; **Navickas, 2019**) in *P. falciparum* (**Supplementary file 2**). To explore whether changes in the NGD pathway are potential adaptations of *P. falciparum* to polyA tracts and poly-lysine repeats, we first used CRISPR/Cas9 technology (**Ghorbal et al., 2014**; **Nasamu et al., 2017**) to HA-tag the endogenous Pelota homolog (PfPelo, PF3D7_0722100, (**Figure 5—figure supplement 1**) in *P. falciparum* Dd2 cells.

Since polyA-containing reporters, as well as endogenous genes, were efficiently and correctly translated in *P. falciparum* cells, we decided to test the *P. falciparum* NGD pathway with a common NGD substrate (**Shoemaker and Green, 2012**; **Doma and Parker, 2006**). We inserted a 78 bp long RNA stem-loop into the fluorescent reporter sequence (StL, **Figure 5—figure supplement 2**). The same stem-loop was previously described to stall ribosomes and induce NGD response with subsequent endonucleolytic cleavage in various reporters and multiple organisms (**Arthur et al., 2015**; **Doma and Parker, 2006**; **Simms et al., 2017**; **Passos et al., 2009**; **Dimitrova et al., 2009**). We followed both protein synthesis and mRNA abundance from the StL construct in *P. falciparum* cells (**Figure 5A and B**). Insertion of the RNA stem-loop resulted in a severe reduction of reporter protein levels (**Figure 5A**, **Figure 5—figure supplement 2**), but moderate increase in mRNA levels (**Figure 5B**). While we could not detect any HA-tagged StL reporter using western blot analyses (**Figure 5A**), we did notice the small but detectable amount of fluorescent reporter in the P. falciparum cells during live imaging (**Figure 5—figure supplement 2**). To test whether stem-loop structure caused ribosomes to stall, we analyzed the distribution of StL transcript in polysome profile (**Figure 5C**). We used constructs with and without polyA tracts as controls. Polysome profile analyses indicated that mRNA of StL construct was associated mostly with polysome fractions 7–9 that correspond to the disome peak. Such mRNA distribution was in sharp contrast with the distribution of mRNAs for constructs with and without polyA tracts ($\pm$polyA$_{36}$) and argues for the potential translational stall (**Figure 5C**). mRNAs for constructs with and without polyA tracts ($\pm$polyA$_{36}$) and for PfGAPDH were distributed more evenly along the polysome fractions (fractions 9–16). These experiments argue that stable RNA structures in mRNAs stall *P. falciparum* ribosomes, reducing protein synthesis from such transcript, however without significant impact on mRNA stability.

To further test whether global arrest of translation leads to the activation of RQC and NGD mechanisms, we used isoleucine (Ile) starvation (**Figure 6A**) to induce widespread ribosome pausing on Ile-codons (**Babbitt et al., 2012**; **Liu et al., 2006**). Parasites rely on extensive proteolysis of human serum proteins to supplement Ile, which is found in 99% of all *P. falciparum* proteins (**Liu et al., 2006**). As such, the majority of ribosomes will have either a reduced rate of protein synthesis or will be stalled on runs of Ile-residues encoded in endogenous *P. falciparum* transcripts. Ile starvation in *P. falciparum* was previously reported to induce a state of hibernation through the arrest of protein synthesis with phosphorylation of eIF2$\alpha$, but in a GCN2-independent fashion, and without existing TOR-nutrient sensing pathway nor activation of ATF4 homologue (**Babbitt et al., 2012**). Surprisingly, during the metabolically induced hibernation, parasites maintain their morphology while slowing down protein synthesis, and importantly more than 50% of parasites recover even after 4 days of starvation (**Babbitt et al., 2012**). By analyzing *P. falciparum* cells 72 hr into Ile-starvation, we found that expression of PfPelo and PfHsp70 chaperone were upregulated in starved cells (**Figure 6B and C**). The mRNA and protein levels of PfPelo were moderately increased (approximately two-fold), while levels of PfHsp70 were strongly induced (five-fold increase) in response to the Ile-starvation. This increase in PfPelo levels is reminiscent of the recently reported increased human Pelo expression during elevated ribosome stalling and the absence of recycling during the developmental transition from reticulocyte to erythrocyte (**Mills et al., 2016**). The substantial increase in Hsp70 levels upon starvation in human cells was associated with cell protective mechanisms against global protein misfolding (**Wu et al., 1985**; **Rosenzweig et al., 2019**).

To test whether Ile-rich transcripts were targeted specifically by the RQC and NGD pathway during Ile-starvation, we examined transcripts that encode Ile-rich proteins in the *P. falciparum* genome. Previous studies indicated that while protein levels did not drastically change throughout starvation, protein synthesis was significantly reduced (**Babbitt et al., 2012**). The microarray analyses of mRNAs in Ile-starvation samples also showed similar overall levels for the majority of mRNA transcripts when compared to non-starved controls. To specifically test transcripts with three and more consecutive Ile-codons, we analyzed mRNA abundance for nine *P. falciparum* genes during Ile-starvation (**Figure 6—figure supplement 1**). We did not observe the general reduction in mRNA levels for Ile-rich *P. falciparum* genes when we compared starved to non-starved control samples. We, instead found

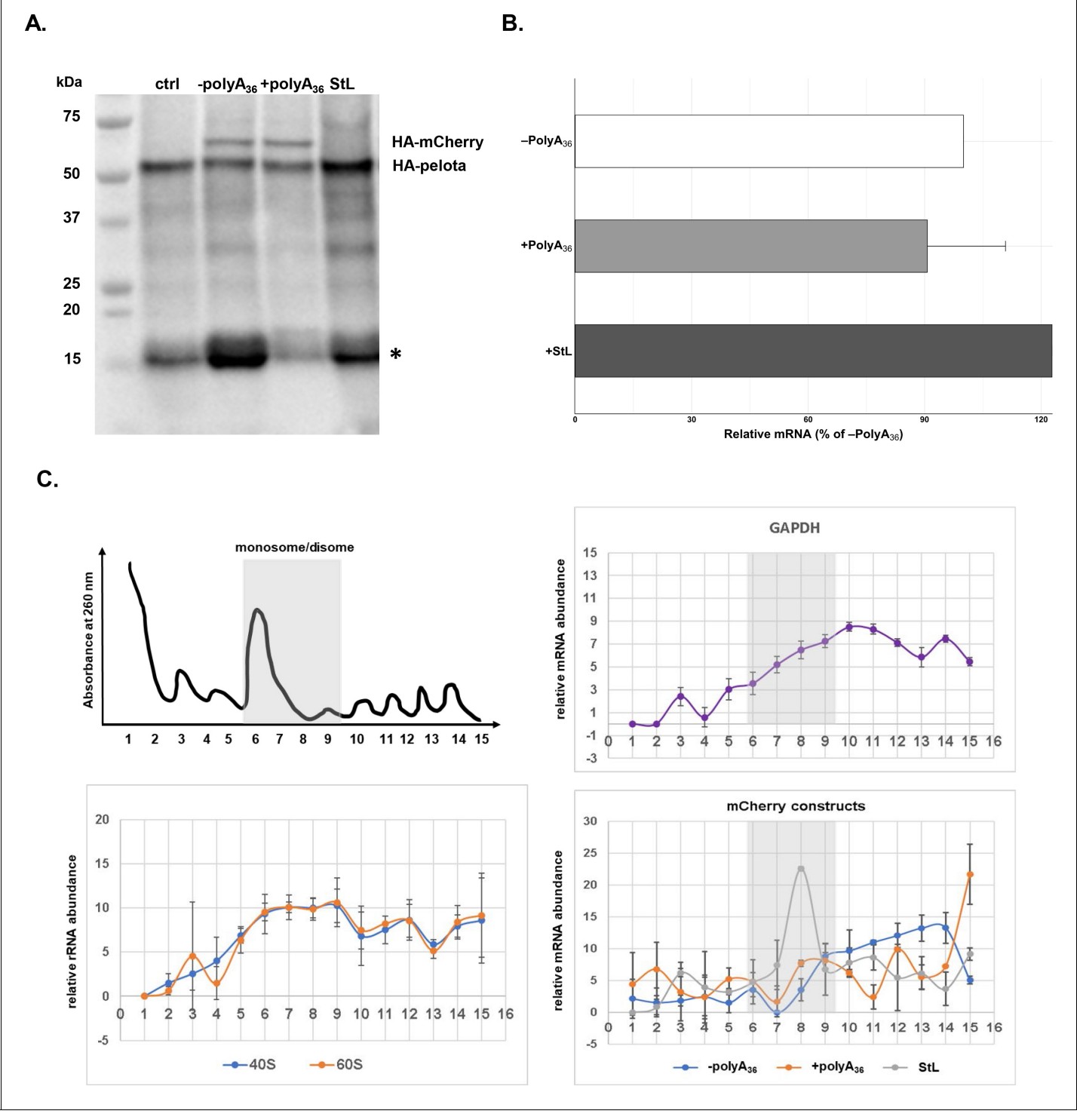

**Figure 5.** mRNA stem-loop pauses P. falciparum ribosome without mRNA degradation. (**A**) The western blot of –polyA36, +polyA36, and stem-loop (StL) tagged reporter gene. The blot was probed with anti-HA antibody HRP labeled (sc-7392HRP). The HA-tagged proteins were episomally expressed in HA-tagged pelota *P. falciparum* Dd2 strain. HA-tagged Pelota protein serves as normalization control. (*) denotes human hemoglobin as 15kD contaminant band appearing and causing cross-reactivity in western blot analyses. (**B**) qRT-PCR measured mRNA levels of of reporter constructs with polyA tract and stem-loop insertions (polyA36, +polyA36, Stl). In each case, data represent the mean value of three biological replicates with standard deviation. (**C**) Distributions of mRNA (GAPDH, -polyA36, +polyA36, Stl) and rRNA, in polysome gradients as determined by RT-qPCR. Error bars represent measurement variability as determined by two qPCR replicates. The position of monosome and disome peaks is indicated in each analyzed sample (gray shade).

*Figure 5 continued*

The online version of this article includes the following source data and figure supplement(s) for figure 5:

**Source data 1.** Polysome profile analyses data.
**Figure supplement 1.** CRISPR/Cas9 HA-tagging of P. falciparum Pelo gene.
**Figure supplement 2.** Schematic of stem-loop reporter construct (StL) used for expression in *P. falciparum*.

a slight increase in mRNA abundance for several Ile-rich transcripts during starvation (*Figure 6C*). This result is rather similar to Leu- and Arg-starvation in mammalian cells, where no overall change in mRNA stability is observed during amino acid starvation (*Darnell et al., 2018*). However, our result indicated that both increase in expression of NGD-associated factor, Pelota, and protein chaperone HSP70 may contribute to cell survival, ribosome recycling and mRNA stabilization during amino acid starvation in *P. falciparum*. This result is opposite to the recently reported role of human Pelo protein in the regulation of global mRNA decay on stalled mRNAs in platelets (*Mills et al., 2017*). Transgenic overexpression of human Pelo protein potentially releases unrecycled ribosomes and stimulates mRNA degradation during translational arrest in platelet differentiation. This is not the case in *P. falciparum*, where the production of the PfPelo protein is increased during the *P. falciparum* Ilestarvation; however, there was a lack of targeted mRNA degradation (*Figure 6C*). We observed, rather, stabilization of mRNAs that should cause ribosomal stalling, like in the case of StL construct (*Figure 5B*). These results may indicate that RQC and NGD pathway in *P. falciparum* cells are not connected, at least directly, to targeted mRNA degradation.

## *Plasmodium* ribosome structure accommodates poly-lysine repeats

In the light of our data with different reporters and analyses of endogenous *P. falciparum* genes (*Figures 2–6*), we were still interested as to how the *P. falciparum* translation machinery deals with polyA tracts and poly-lysine sequences. Ribosomal stalling on long poly-lysine runs (*Arthur et al., 2015*; *Juszkiewicz and Hegde, 2017*; *Sundaramoorthy et al., 2017*; *Dimitrova et al., 2009*; *Kuroha et al., 2010*; *Brandman et al., 2012*; *Lu and Deutsch, 2008*) was classically explained by electrostatic interactions of the polybasic peptide and the exit tunnel of the ribosome (*Lu and Deutsch, 2008*). More recent studies revealed that polyA tracts and mRNA directly contribute to ribosomal stalling and frameshifting (*Arthur et al., 2015*; *Koutmou et al., 2015*; *Chandrasekaran et al., 2019*; *Tesina et al., 2020*). Consecutive adenosines are engaged by the ribosome decoding center nucleotides and are stabilized on both sides by rRNA base stacking interactions (*Chandrasekaran et al., 2019*; *Tesina et al., 2020*), and adopt a helical conformation typical for single-stranded polyA stretches (*Tang et al., 2019*). Based on these reports, we analyzed *P. falciparum* ribosomes as the principal components that could accommodate translation of long polyA stretches.

The retention and conservation of polyA tracts, as well as the stage-independent expression of polyA genes (*Figure 1*), reveals that all rRNAs within *Plasmodium spp.* must deal with them. The structural data on *Plasmodium* ribosomes are limited to two recent cryo-EM studies of *P. falciparum* ribosomes isolated during the schizont stage of the IDC (*Wong et al., 2014*; *Sun et al., 2015*). Both studies reported that *P. falciparum* ribosomes have different structural and dynamic features that distinguish them from other organisms. While the majority of *P. falciparum* 28S associated rRNA expansion segments (ESs) are shorter than in human rRNA, 18S ESs are generally larger (*Figure 7— figure supplement 1*). The ES6S/7S is located next to the binding pocket of eIF3 and might be necessary for translation initiation or termination (*Wong et al., 2014*; *Beznosková et al., 2013*; *Hashem et al., 2013*). ES9S/10S are positioned at the head of the 40S subunit and are probably important for the recruitment of additional translation factors as well as for mobility of the head region of 40S subunit (*Wong et al., 2014*; *Sun et al., 2015*). Previous studies had shown that the absence of ZNF598 and RACK1 would help in the resolution of putative stalling and subsequent read through of polyA sequences (*Garzia et al., 2017*; *Juszkiewicz and Hegde, 2017*; *Sundaramoorthy et al., 2017*). However, the *P. falciparum* homolog of ZNF598 (PF3D7_1450400), appears to be abundantly expressed according to PaxDB database (*Wang et al., 2015*).

On the other hand, while being essential and one of the most well-expressed proteins in *P. falciparum* cells (*Wang et al., 2015*; *Blomqvist et al., 2017*), the absence of RACK1 protein on

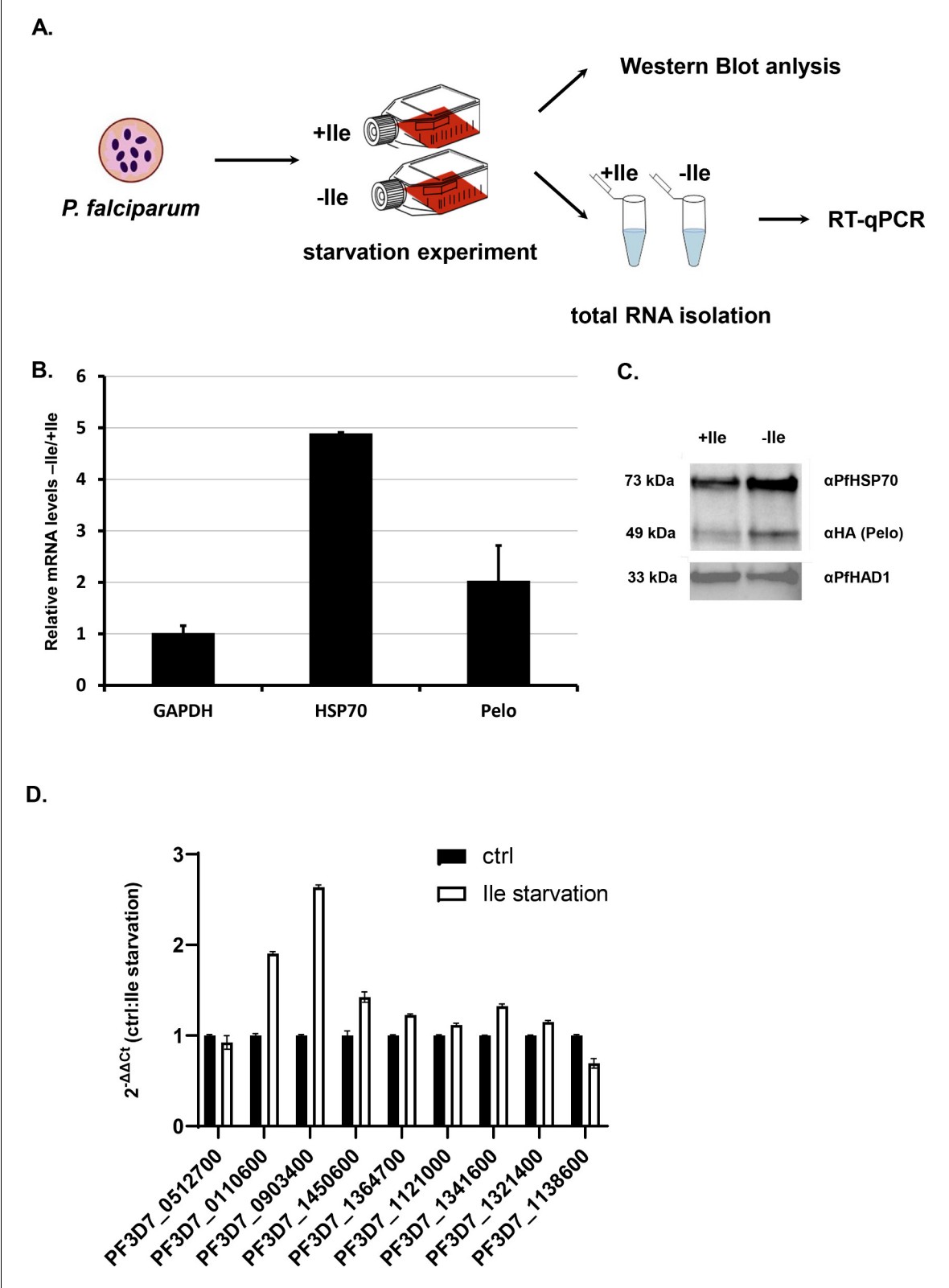

**Figure 6.** Isoluecine starvation in Plasmodium cells induces HSP70 and Pelo genes. (A) The schematic presentation of the starvation experiment design. The cells were incubated for 48 hr in medium with and without Ile (+ / - Ile). After 48 hr samples were collected and protein and total RNA was isolated. (B) Relative mRNA of *P. falciparum* GAPDH (PF3D7_1462800), HSP70 (PF3D7_0818900), Pelo (PF3D7_0722100) transcripts after 48 hr isoleucine (Ile) starvation of the *P. falciparum* cells. Levels of each transcript are normalized to total GAPDH levels and represented as a ratio of transcript levels under

*Figure 6 continued on next page*

*Figure 6 continued*

Ile starvation (-Ile) over the control conditions (+Ile). In each case, data represent the mean value of three biological replicates with standard deviation. (C) Levels of *P. falciparum* HSP70 and Pelo proteins after 48 hr of Ile starvation. Western blot analysis of Ile starved (-Ile), and control (+Ile) sample are normalized to PfHAD1 (PF3D7_1033400) levels. HA-tagged *P. falciparum* Pelo protein was CRISPR/Cas9 engineered and detected using mouse HA-antibody. The molecular weight of each protein is indicated. (D) qRT-PCR measured mRNA levels of genes containing 3–7 Ile. PF3D7_0322300, PF3D7_0512700, PF3D7_0110600, PF3D7_0903400, PF3D7_1450600, PF3D7_1364700, PF3D7_1121000, PF3D7_1341600, PF3D7_1321400, PF3D7_1138600. Levels of each transcript are normalized to total GAPDH level and represented as a ratio of transcript levels under Ile starvation (-Ile) over the control conditions (+Ile). In each case, data represent the mean value of three biological replicates with standard deviation.

The online version of this article includes the following source data and figure supplement(s) for figure 6:

**Source data 1.** qRT-PCR data for Ile-rich transcripts.

**Figure supplement 1.** Schematics of nine *P. falciparum* genes with runs of 3 and more consecutive isoleucine (Ile) residues used in analyses shown in *Figure 6D*.

---

*Plasmodium* ribosomes could be the most prominent features regarding the translation of polyA tracts and poly-lysine repeats (*Wong et al., 2014*; *Sun et al., 2015*; *Figure 7—figure supplement 1*). Asc1 (yeast RACK1 homolog) is required for endonucleolytic cleavage of stalled mRNAs in yeast cells (*Matsuo et al., 2017*; *Ikeuchi and Inada, 2016*). The deletion of RACK1/Asc1 in human or yeast cells was also shown to increase the production of proteins with polybasic peptides, however, at the cost of increased frameshifting (*Sundaramoorthy et al., 2017*; *Kuroha et al., 2010*; *Wolf and Gray-hack, 2015*). The recent report has shown that the absence of Asc1 in yeast generally slows down elongation, promoting frameshifting on problematic CGA-CGA pairs (*Tesina et al., 2020*). However, the mechanisms proposed for the polyA-induced ribosome stalling involves the inhibitory conformation of polyA tract mRNA in the A site of the ribosome (*Chandrasekaran et al., 2019*; *Tesina et al., 2020*). This proposed mechanism for polyA stalling induction should not depend on species, because ribosome nucleotides interacting with π-stacked polyA sequences are universally conserved (A1756 and C1634, *Figure 7—figure supplement 2*). No significant differences in the mRNA tunnel have been observed in *P. falciparum* ribosome structures (*Wong et al., 2014*; *Sun et al., 2015*). However, differences in the resolution and absence of mRNA in *P. falciparum* ribosome cryo-EM structures do not allow for precise comparison of the conformation of rRNA between *Plasmodium* and either yeast or mammalian ribosomes (*Wong et al., 2014*; *Sun et al., 2015*). Altogether these observations do not explain why the mRNA tunnel and interactors of the *P. falciparum* ribosome could influence the efficient translation of polyA tracts.

Recently it was postulated that the stalling on the polyA tracts is an effect of a synergy between polyA-induced suboptimal geometry at PTC and poly-lysine interactions with ribosome exit tunnel (*Chandrasekaran et al., 2019*).To further investigate whether additional differences in the ribosome structure contributed to *P. falciparum* adaptation to long polyA tracts and poly-lysine repeats, we analyzed the ribosome peptide exit tunnel of *P. falciparum* as predicted by the MOLE 2.5 toolkit (*Figure 7*; *Sehnal et al., 2013*). While a recent study found no differences in charge pattern along the peptide tunnel in eukaryotes, notably between *P. falciparum* and human (*Dao Duc et al., 2019*), we observed changes in hydrophobicity across the tunnel (*Figure 7* and *Figure 7—figure supplement 3*). Interactions between clusters of positive charges and hydrophobic environments are generally unfavorable even during protein translocation (*Fujita et al., 2011*). Along this line, the ribosome exit tunnel of *P. falciparum* seems to be generally more hydrophilic, including the constriction site. Given a substantial length of poly-lysine peptide required to induce stalling on downstream polyA tracts (6 to 11, depending on the experimental system) (*Chandrasekaran et al., 2019*) it is plausible that the upper and the central tunnel are contributing to the observed effects. This is what is observed in other organisms. The free energy profile of lysine across the *H. marismortui* ribosome exit tunnel, which also has a hydrophobic region between the PTC and constriction site (*Figure 7—figure supplement 3*), indicates the presence of a significant energy barrier after the constriction site, not before (*Petrone et al., 2008*). Indeed, in vitro translation of poly-lysine and poly-phenylalanine using *E. coli* ribosomes resulted in rather different paths and rates of the extension of nascent peptide due to postulated hydrophobic entry to the ribosome exit tunnel (*Picking et al., 1991*). Moreover, in a recent structure of the polyA stalled yeast ribosome, the unresolved density was also indicated in this part of the exit tunnel next to the PTC and above the constriction site (*Tesina et al., 2020*). Taken together, it seems that the ribosome peptide exit tunnel of parasites adapted to

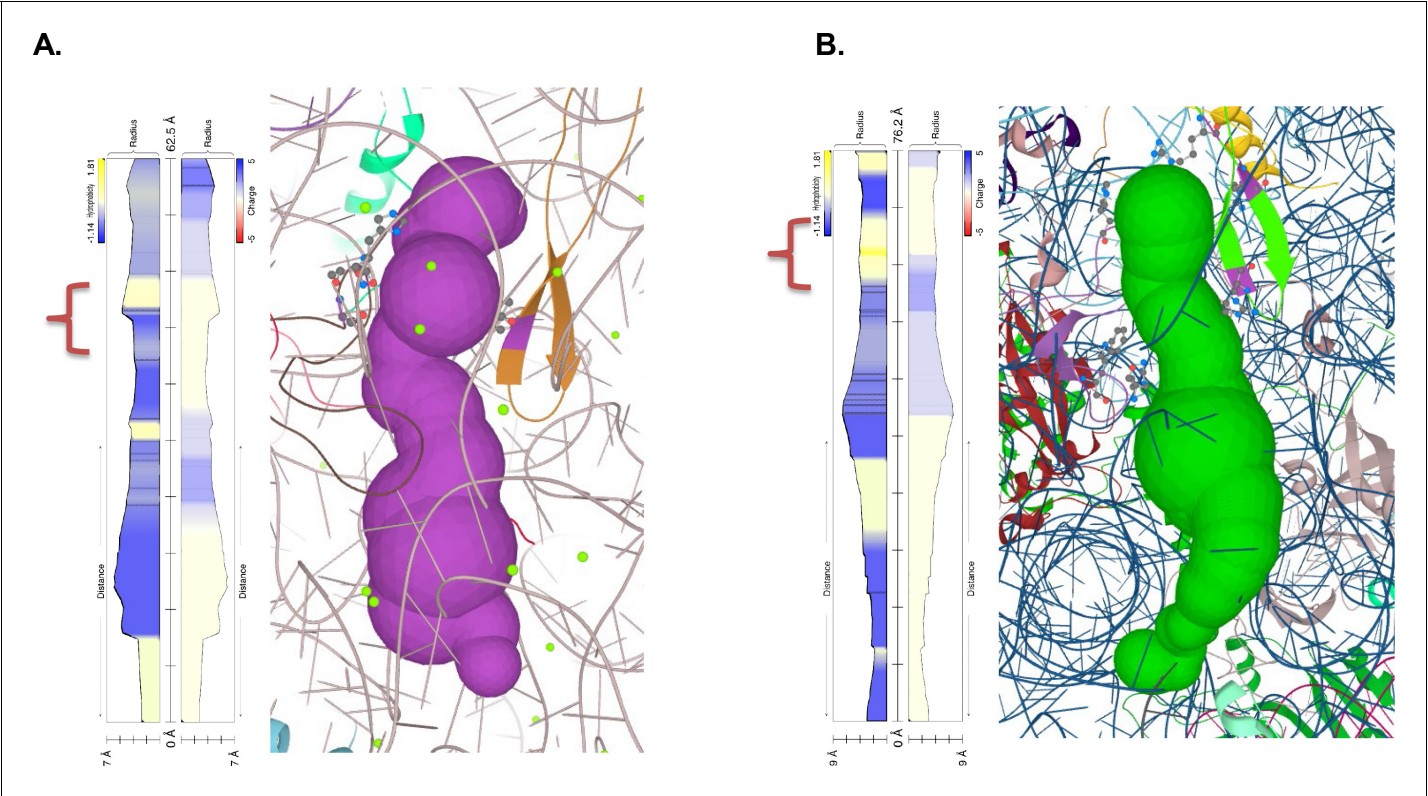

**Figure 7.** Polypeptide exit tunnel as predicted by MOLE service from *Plasmodium* ribosome (PDB:3j79) (**A**) and human ribosome (PDB: 6d90) (**B**). The constriction site flanked by L22 and L4 is marked in the orange clamp, and lining residues in both cases are marked with ball-and-stick visualization. Vertical plot outline hydrophobicity (left) and charge (right) profiles across the tunnel (*Pravda et al., 2018*). The width of the tunnel is indicated on the scale at the bottom of the plot. The total lengths of the tunnel are indicated at the top of the plots.

The online version of this article includes the following figure supplement(s) for figure 7:

**Figure supplement 1.** Structure of *P. falciparum* (PDB code: 3JBO) and *H. sapiens* (PDB code: 3JAG) ribosomes with receptor for activated kinase C (RACK1) in magenta, previously shown to be absent from *Plasmodium* ribosomes (60S in green, 40S in cyan).

**Figure supplement 2.** Segment of sequence alignment of 18S rRNA from *P. falciparum* (Pf18SA and Pf18SS, A- and S-type of ribosomes), *H. sapiens* (Hs18S), *S. cerevisiae* (Sc18S), *T. thermophila* (Tt18S) and *E. coli* 16S (Ec16S).

**Figure supplement 3.** Polypeptide exit channel from Haloarcula marismortui ribosome (PDB: 1jj2) has one long fragment of relatively hydrophobic lining of the tunnel at the entrance (between PTC and constriction site).

accommodate synthesis of long runs of poly-lysine residues by reducing electrostatic barriers below the PTC and at the constriction site. The resulting dynamics of translation of poly-lysine might be high enough to reduce the chances for unfavorable interactions between mRNA and ribosome.

## Discussion

Runs of lysine and arginine residues are underrepresented in the proteome of multiple organisms compared to the runs of other amino acids, suggesting a selective pressure against polybasic amino acid sequences (*Karlin et al., 2002*). One potential explanation for this evolutionary selection is electrostatic interactions of the polybasic peptide and the exit tunnel of the ribosome that would impact the rate of protein synthesis (*Brandman et al., 2012*; *Lu and Deutsch, 2008*). However, multiple biochemical analyses of poly-lysine stalling sequences, in both prokaryotic and eukaryotic systems, revealed that runs of lysine AAA codons exhibit a more significant delay in ribosome movement than an equivalent number of AAG lysine codons (*Glöckner, 2000*; *Szafranski et al., 2005*; *Zilversmit et al., 2010*; *Arthur et al., 2015*; *Koutmou et al., 2015*; *Tournu et al., 2019*; *Szádeczky-Kardoss et al., 2018*; *Chandrasekaran et al., 2019*; *Tuck et al., 2020*; *Tesina et al., 2020*). More recent study in mouse embryonic stem cells indicates that besides 12A-1 sequences endogenous sites with 8–9 As are sufficient to cause recruitment of mRNA surveillance pathways

(*Tuck et al., 2020*). So as is the case of polybasic residues, analyses of codon usage within poly-lysine peptides revealed that the expected frequency of AAA codons compared to AAG codons in runs of four consecutive lysine residues is lower in the coding regions of genes for multiple organisms (*Arthur et al., 2015*; *Koutmou et al., 2015*; *Habich et al., 2016*; *Figure 1*). However, ribosome stalling and frameshifting induced by polyA runs depend on the sequence and conformation of the mRNA in the ribosome, with a questionable contribution by the nascent polypeptide chain (*Arthur et al., 2015*; *Arthur and Djuranovic, 2018*; *Koutmou et al., 2015*; *Tesina et al., 2020*). As such, both polyA tracts and poly-lysine sequences are unfavorable in mRNAs or proteins, respectively, of most tested organisms.

Data presented here indicate *P. falciparum,* the most studied malaria-causing human pathogen, is an exception to this rule. With an 80% AT-rich genome and more than 63% of transcripts containing polyA tracts, *P. falciparum* represents a paradigm breaking species for polyA tracts (*Figure 1*). The atypical enrichment in polyA tracts in the coding sequence of mRNAs, translated into long poly-lysine runs in proteins, is a feature that is conserved in most *Plasmodium* species (*Figure 1*; *Caro et al., 2014*; *Habich et al., 2016*; *Wang et al., 2015*; *Aurrecoechea et al., 2009*). Such species-specific evolutionary conservations usually result in traits that are beneficial for survival. The increased AT-richness and number of polyA tracts in *P. falciparum*, as such, could be a result of selective pressure on biosynthetic pathways (AT vs. GC biosynthesis) (*Seward and Kelly, 2016*; *Dietel et al., 2019*) and the oxidative intraerythrocytic environment (*Becker et al., 2004*). An increase in poly-lysine runs might have been of different etiology. Gene ontology analyses of polyA tract and poly-lysine genes in *P. falciparum* suggested cellular and pathological adhesion proteins as one of the enriched gene groups (*Supplementary file 1*). It is attractive to speculate that advantages for the parasites to synthesize proteins with poly-lysine repeats could be solely driven by benefits in the adhesion and invasion of host cells (*Kobayashi et al., 2013*; *Leitgeb et al., 2011*) or export of numerous parasite proteins to the surface of infected erythrocytes (*Davies et al., 2016*; *Romero et al., 2004*; *Hancock et al., 1990*).

Our data further indicate that the *P. falciparum* translational machinery permits the translation of polyA tracts and poly-lysine repeats without mRNA degradation, protein synthesis attenuation, nor obvious activation of translational surveillance pathways (*Figures 2–5*). The biochemical assays with reporter sequences (*Figures 2*, *3* and *5*) as well as analyses of ribosome profiling data (*Caro et al., 2014*; *Figure 4*) suggest that parasite ribosomes do not stall or frameshift on long polyA tracts and poly-lysine runs. Even though RNA folding propensity in *P. falciparum* has been skewed due to the nucleotide content, the predicted stability of native *P. falciparum* transcripts did not show differences compared to human cells. However, the change in nucleotide composition and a number of long polyA tracts of *P. falciparum* (*Figure 1*) probably shaped NGD/NSD mRNA surveillance responses in parasites (seen in *Figures 5* and *6*). Insertion of the stable stem-loop structure (StL) in the reporter construct resulted in reduced protein synthesis and ribosome stalling but without noticeable mRNA degradation (*Figure 5* and *Figure 5—figure supplement 2*). Our results with stem-loop construct and Ile-starvation are rather disparate to experiments in other eukaryotes, including the human host (*Arthur et al., 2015*; *Doma and Parker, 2006*; *Simms et al., 2017*; *Passos et al., 2009*; *Dimitrova et al., 2009*), and argues for significant differences between human and *Plasmodium* NGD mRNA surveillance pathways.

Similarly, while following the general cellular response to induced metabolic stress with Ile-starvation we noticed the upregulation of Pelo and Hsp70 levels (*Mills et al., 2016*; *Wu et al., 1985*; *Rosenzweig et al., 2019*), however again with the lack of targeted mRNA degradation of Ile-rich transcripts or reduction in global mRNA stability (*Babbitt et al., 2012*; *Figure 6*). It seems plausible that amino acid starvation could allow uncharged Ile-tRNAs to fill the A-site and block Pelo binding thus preventing NGD and RQC activation in this case. Similarly, Leu- or Arg-starvation experiments in HEK293 cells do not result in global change in mRNA stability but rather increase in translation termination and ribosome recycling (*Darnell et al., 2018*). While authors in Hek293 amino acid starvation studies did not follow changes in Pelota protein levels, it was shown recently that transgenic overexpression of Pelo protein in human platelets led to rapid degradation of most cytoplasmic platelet transcripts (*Mills et al., 2017*). Our studies indicate that increased *P. falciparum* Pelo levels, at least during the amino acid starvation, do not cause targeted mRNA degradation. However, future studies on the role *Pf*Pelo levels during amino acid starvation, its role in ribosome recycling as well as identification of missing *P. falciparum* Hbs1 protein will give more insight in NGD and RQC

process in *Plasmodium*. Nonetheless, results with the Stl construct and amino acid starvation suggest a difference between human and *Plasmodium* NGD mRNA surveillance and mRNA degradation pathways in both signaling and resolution of ribosome stalling events (*Joazeiro, 2017*). One of these differences could be potentially attributed to the loss of recently reported NGD endonucleases (*D'Orazio et al., 2019*; *Glover, 2019*; *Navickas, 2019*) in *P. falciparum* cells. The bioinformatics analyses of the *P. falciparum* genome did not indicate a homolog of either Cue2 or Nonu-1 endonuclease (*Supplementary file 2*; *D'Orazio et al., 2019*; *Glover, 2019*; *Navickas, 2019*). However, the loss of Cue2/Nonu-1 in *Plasmodium* cells can not completely explain the lack of mRNA degradation on the stem-loop construct (*Figure 5*) or in Ile-starvation experiments (*Figure 6*). Given that the majority of observed NGD-induced mRNA decay is processed by canonical exonucleolytic decay by Xrn1 (*D'Orazio et al., 2019*; *Navickas, 2019*), stabilization of NGD substrates in *P. falciparum* remains puzzling.

It is apparent that the accommodation of polyA tracts and poly-lysine repeats in translation required multiple adaptations to be vital and typically highly conserved components of the translation and mRNA surveillance pathways. Unambiguously, *P. falciparum* ribosomes had to change to accommodate correct and efficient protein synthesis from the runs of coding polyA tracts that are longer than the average size of 3'-UTR polyA tails in other organisms (*Subtelny et al., 2014*; *Brown and Sachs, 1998*; *Chang et al., 2014*). The fact that recent cryo-EM structures indicated that the *Plasmodium* ribosome lacks interaction with PfRACK1 protein (*Figure 7*) could be beneficial for the translation of polyA tracts into poly-lysine runs (*Wong et al., 2014*; *Sun et al., 2015*). Albeit, it is not clear what would substitute the role of RACK1/Asc1 in the correct reading frame maintenance during translation of long polyA tract or other stalling sequences (*Sundaramoorthy et al., 2017*; *Tesina et al., 2020*; *Kuroha et al., 2010*; *Wolf and Grayhack, 2015*). The lack of Asc1/RACK1 association with *P. falciparum* ribosomes could affect its association with other RQC protein homologs, such as ZNF598, which can influence the activation of RQC pathway and endonucleolytic cleavage (*Ikeuchi and Inada, 2016*). It is possible to speculate that *P. falciparum* specific ribosome extension segments in the 40S subunit may have the role in recruiting other *Plasmodium* proteins that may help with translation of polyA tracts and poly-lysine repeats (*Wong et al., 2014*; *Sun et al., 2015*). However, it was recently indicated that an inhibitory conformation of mRNA in the A-site of the ribosome, as well as the polybasic character of the nascent polypeptide chain in the ribosome, peptide exit tunnel, are crucial for the poly(A)-mediated stalling mechanism (*Chandrasekaran et al., 2019*; *Tesina et al., 2020*). It is not clear from the conservation of the 18S rRNA sequence (*Figure 7—figure supplement 2*) and structure of the ribosome A-site in *P. falciparum* ribosomes (*Wong et al., 2014*; *Sun et al., 2015*) how parasite ribosomes would circumvent an inhibitory conformation of the polyA mRNA in the A site seen in yeast and mammalian ribosomes (*Chandrasekaran et al., 2019*; *Tesina et al., 2020*). It might be that changes in the peptide exit tunnel of *P. falciparum* ribosomes (*Figure 7*) have partially contributed to the adaptation to the synthesis of long poly-lysine runs by reducing the potential stalling effects associated with translocation of polybasic peptides through hydrophobic environment (*Fujita et al., 2011*). However, contributions of *Plasmodium*'s ribosome exit tunnel in overcoming poly-lysine induced ribosome stalling still need to be tested experimentally. It is also plausible that translation rate is a factor here. Chandrasekaran and coworkers (*Chandrasekaran et al., 2019*) in their polyA stalling study had observed a significant difference in the number of lysines required for stalling between in vitro system and cells. They have explained the observation by a 5-fold differences in translation rate.

In conclusion, to adapt to polyA tract translation for the production of the polybasic and homopolymeric lysine repeats, the malaria-causing parasite has altered its mRNA translation quality control pathways as well as its ribosomal proteins and ribosome structure. The additional diversity of *Plasmodium spp.* rRNAs (*Walliker et al., 1987*; *McCutchan et al., 1988*; *Waters et al., 1989*) and possible activity of yet unknown ribosome-associated factors promote the possibility of *Plasmodium* 'specialized ribosomes,' which allow for polyA tract translation into poly-lysine sequences of functional proteins. Further insights into differences between components of translational machinery and mRNA surveillance pathways present in *P. falciparum* and host organisms, as well as the physiological role(s) of conserved poly-lysine repeats, will provide answers to these questions and enable identification of new drug targets against malaria.

# Materials and methods

## Key resources table

| Reagent type (species) or resource | Designation | Source or reference | Identifiers | Additional information |
|---|---|---|---|---|
| Gene (*Plasmodium falciparum*) | Pelota gene | This paper | PF3D7_0722100 | CRISPR/Cas9 engineered with C-terminal 3xHA tag |
| Cell line (*Plasmodium falciparum*) | *Dd2 strain* | Goldberg lab | http://neurolex.com/wiki/NCBITaxon:57267 | *P. falciparum* strain used for all experiments |
| Cell line (*Homo sapiens*) | Dermal fibroblast (normal, Adult) | ATCC | PCS-201–012 | |
| Cell line (*Homo sapiens*) | Adult erythrocytes (primary cell line) | Goldberg lab and BJCIH Children Hospital | | Deidentified human blood. 50% haematocrit, washed in complete medium (MCM) |
| Cell line (*Tetrahymena thermophila*) | strain B2086 (II) | Chalker Lab | http://neurolex.com/wiki/NCBITaxon:5911 | |
| Recombinant DNA reagent | pBSICY-gtw (plasmid) | Chalker Lab | | • PMCID:PMC3232721 • *T. thermophila* transfection plasmid |
| Recombinant DNA reagent | pc-DNA-DEST40 (plasmid) | Thermo Fisher Scientific | RRID:Addgene_45597 | HDF transfection plasmid |
| Recombinant DNA reagent | pHHT-TK (plasmid) | Goldberg Lab | | Plasmid for standard CRISPR/Cas9 knock out approach in *P. falciparum* cells. |
| Recombinant DNA reagent | pyDHOD-2A-Cas9 (plasmid) | Goldberg lab | | All-in-one Cas9+gRNA cassette vector DSM-1 resistance |
| Antibody | anti-HA antibody (mouse monoclonal antibody) | Santa Cruz Biotechnology | Cat# sc-7392, RRID:AB_627809 | WB (1:5000), used also as HRP conjugated |
| Antibody | anti-mouse (horse unknown clonality) | Cell Signalling Technology | Cat# 7076, RRID:AB_330924) | WB (1:5000) |
| Antibody | anti-rabbit (goat polyclonal antibody) | Cell Signalling Technology | Cat# 7074, RRID:AB_2099233 | WB (1:5000) |
| Antibody | Anti-HSP70 (rabbit polyclonal antibody) | AgriSera | Cat# AS08 371, RRID:AB_2248616 | WB (1:1000) |
| Antibody | anti-PfHAD1 rabbit | Odom lab | | WB (1:10000) PMID:25058848 |
| Chemical compound, drug | Cycloheximide | Sigma Aldrich | Cat# C1988 | 200 uM |
| Chemical compound, drug | WR99210 | Sigma Aldrich | Cat# W1770 | 10 nM |
| Chemical compound, drug | DSM1 | Sigma Aldrich | Cat#533304 | DHODH Inhibitor 1.5 uM |
| Commercial assay or kit | llustra triplePrep Kit | GE Healthcare | Cat# 28942544 | DNA, RNA and protein isolation kit |

## Parasite culturing

*P. falciparum* line Dd2 was cultured at 2–5% hematocrit in $O^+$ erythrocytes in RPMI 1640 supplemented with 5 g/l Albumax II (Gibco), 0.12 mM hypoxanthine (1.2 ml 0.1M hypoxanthine in 1 M NaOH), 10 µg/mL gentamicin (*Trager and Jensen, 2005*). Cultures were grown statically in a candle jar atmosphere. As required, cultures were synchronized with 5% (wt/vol) sorbitol. Medium for Isoleucine starvation: 16.1 g RPMI medium 1640 per l (ME011232P1 Gibco), 2 g of sodium bicarbonate,0.12 mM hypoxanthine, 10 µg/mL gentamicin, and with and w/o 100 uM Isoleucine.

Generation of mCherry-GFP–Expressing plasmids with and without 12 lysines coded with 36 As in a row (polyA- and polyA+) mCherry with and without polyA was amplified from plasmids previously created (*Arthur et al., 2015*) pBttlysforXhoI 5′-gcgcctcgagatgggctacccatacga-3′ (XhoI site underlined) and mCherryrevAvrII 5′-gcgccctaggcttgtacagctcgtccatgccg3′ (AvrII site highlighted) digested with XhoI and AvrII, and ligated into the same sites of the episomal over-expression (EOE) a containing the promoter region for PfHsp86, a C-terminal GFP tag, and a human dihydrofolate reductase (hDHFR) drug-selection cassette (*Russo et al., 2009*). mCherry reporter in pc-DNA-DEST40 previously published *Arthur et al., 2015*; *Arthur et al., 2017* used to transfect HDF cells.

## HDF cells culturing and transfection

HDF cells (ATCC PCS-201–012) were cultured in Dulbecco's modified Eagle's medium (DMEM) (Gibco) and supplemented with 10% fetal bovine serum, 5% minimum essential medium nonessential amino acids (100×, Gibco), 5% penicillin and streptomycin (Gibco), and L-glutamine (Gibco). HDF cells were Mycoplasm-free based on PCR-tests using Venor GeM Mycoplasma Detection Kit (Sigma Aldrich).

mCherry reporter in pc-DNA-DEST40 plasmid (1 ug) was introduced to the cells by the Neon Transfection System (Invitrogen) with 100 µl tips according to cell-specific protocols (www.lifetechnologies.com/us/en/home/life-science/cell-culture/transfection/transfection—selection-misc/neon-transfection-system/neon-protocols-cell-line-data.html). Cells electroporated with DNA plasmids were harvested after 48 hr if not indicated differently (*Arthur et al., 2015*).

## Parasite transfection

Asynchronous Dd2 parasites were incubated with 100 µg of maxi-prep DNA of each of the EOE constructs described above encoding mCherry-GFP (with and w/o polyA), transfected in Bio-Rad Gene Pulser cuvette (0.2 cm), 0.31 kV, 950 up, infinity resistance. 10 nM WR99210 was added to parasite 48 hr after transfection and used to select resistant parasites (*Fidock and Wellems, 1997*).

## Saponin lysis of infected red blood cells (iRBC)

The cell iRBCs were resuspended in two volumes of PBS containing 0.15% saponin, and incubated on ice for 10 min, with vigorous mixing every 3 min. Afterward, the samples were centrifuged 7000 g, 5 min, 4°C, and the pellets were washed three times more with the same buffer.

## Genomic DNA (gDNA) extraction

The extraction of gDNA was done from samples pretreated with a saponin lysis buffer, and we followed the protocol of the DNeasy Qiagen kit.

## RNA extraction and qRT-PCR

Total RNA was extracted from iRBCs using the RiboZol RNA extraction reagent (Amresco) with some changes (protocol from Jabos-Lorrrena Laboratory at Johns Hopkins School of Public Health), or we used Illustra triplePrep Kit GE Health Care. In 1 ml of pelleted iRBCs (500 g, 5 min, room temperature), parasitemia 5–7%, 5 ml of RiboZol was added. It was mixed thoroughly that all cells lysed. At this step, the sample can be snap-frozen and stored at −80°C or proceed further. Add 1 ml of chloroform and mix for 20 s. Transfer the aqueous phase to a new tube, and add 1/10th of the volume of 3 M sodium acetate, mix well and add 10 µl GlycoBlue. Add 2.5 ml of cold iso-propanol and leave overnight at −80°C. After overnight precipitation, the samples were centrifuged at 16000xg, 1 hr, 4°C. The supernatant was discarded and washed twice with 5 and 1 ml of cold ethanol. The pellets were dried and resuspended in pure water. RNA concentration was measured by NanoDrop (OD260/280). For total RNA extraction using Illustra TriplePrep Kit (item# 28942544 GE Health Care), we used 200 µl of iRBCs, parasitemia 5–7%. The cell pellet was resuspended in 600 µl PBS buffer containing 0.15% saponin (item# 47036–50 G-F Sigma), cOmplete Protease Inhibitor Cocktail without EDTA (item# 11873580001 Roche), RNase OUT 1 µl/ml (item# 10777019 Invitrogen), and incubated at RT for 10 min on ice. Afterward, it was centrifuged 7000xg, 5 min, 4°C, and the pellets were washed three times more with the same buffer. After this pretreatment to get rid of hemoglobin pellets were used for total RNA isolation following Illustra TriplePrep Kit protocol. iScript Adv cDNAkit for RT-qPCR (item# 172–5037, Bio-Rad) or SuperScript *IV VILO* Master Mix(item# 11756050

Life Technologies Corporation), was used with 100–200 ng of total RNA following the manufacturer's protocol. iQ SYBR Green Supermix (item# 1708886, Bio-Rad) protocol was used for qRT-PCR on the CFX96 Real-Time system with Bio-Rad CFX Manager 3.0 software (*Arthur et al., 2015*). Cycle threshold ($C_t$) values were normalized to the hDHFR resistance gene expressed from the same plasmid or HSP86, GAPDH genes.

## qRT-PCR primers

**PF3D7_14628002GAPDHqf**
5'-ACCAAAGGATGACACCCCAA
2GAPDHqf 5'-ACCACCCTTTGATGGACCAT
**PF3D7_1464200**
610gf 5'-CGACAAGGCCATTTTAGAGAA-3'
610gR 5'TTTCGTTTTATCTCCGCTTACA-3'
**PF3D7_0615500**
0615500for 5'-CCACAATTGGAGTCGTCGTA-3'
0615500rev 5'-TCAAATCGAATTCTGTGACTCCT-3'
**UniProtKB - P00374**
qPCR_ hDHFR_F_5'-TCCTCCTGGACATCAGAGAGA-3' qPCR_ hDHFR_R_5'-CTCAAGGAACCTCCACAAGG-3'
**PF3D7_0818900**
Hsp70qf 5'-GAATCGGTTTGTGCTCCAAT-3'
Hsp70qr 5'-CAACTGTTGGTCCACTTCCA-3'
**PF3D7_1108000**
qPCRrevlWS1w 5'-TGGTTGAAGAGGATGAGGAGA-3' qPCRrevlWS1w 5'-ACCTTGTGCATATCATCATTTTTCC-3' mCherry mCherry qF 5'-TGACGTACCGGATTATGCAA-3' mCherry qR 5'-ATATGAACTGAGGGGACAGG-3'
**Ile rich genes**
PF3D7_0322300_F 5'-TGGATGATCTGAGCAACAAAA-3'
PF3D7_0322300_R 5'-GGGTGGATCTTTATGCAAGC-3'
PF3D7_0512700_F 5'-TGGAACAGCATTAACGGAAA-3'
PF3D7_0512700_R 5'-GAGGTATTCCTACCCTTTTCTCAA-3'
PF3D7_0110600_F 5'-AGCATCACGACCTTTCCATC-3'
PF3D7_0110600_R 5'-TTGCATAAGCATTGGGATGA-3'
PF3D7_0903400_F 5'-TTCCATTATTGCATGCTCTCC-3'
PF3D7_0903400_R 5'-TCACACATGGATGTTGCTCA-3'
PF3D7_1450600_F 5'-ACGGATTACATGCAGCACAA-3'
PF3D7_1450600_R 5'-GATGACGTGTCGTCAAAAA-3'
PF3D7_1364700_F 5'-AAGGAAGCTCGGTTTTATTTGA-3'
PF3D7_1364700_R 5'-AAACCCTTCTTTTGTTTTGACA-3'
PF3D7_1121000_F 5'-CAAAAACAAATCCCGTAGATCC-3'
PF3D7_1121000_R 5'-CGATACAATTGTTGACCCACAT-3'
PF3D7_1341600_F 5'-GGGAATGGGAACCTTGTGTA-3'
PF3D7_1341600_R 5'-TCTTCATTTATCCATGCGTCA-3'
PF3D7_1321400_F 5'-TCCTTTCCATCCTCCCTTTT-3'
PF3D7_1321400_R 5'-TGGATTTTATCCACGGGTGT-3'
PF3D7_1138600_F 5'-ACAAGCGGAAAATATCGAATG-3'
PF3D7_1138600_R 5'-TCGTCTAAGTCCACTTCACTGC-3'

## Immunoblotting and antibodies

Samples pretreated with lysis buffer supplemented with cOmplete Protease Inhibitor Cocktail without EDTA were prepared with passive lysis buffer (Promega), BioRad sample buffer and BioRad reducing buffer. For immunoblotting, the PVDF membranes were blocked in 5% milk in PBS. The membranes were probed with anti-HA, or Anti-HA HRP (sc-7392, sc-7392HRP, Santa Cruz) or anti-HA mouse or rabbit antibody (7076 s, 7074 s respectively, Cell Signaling) diluted in 2.5% milk in PBS-Tween20 0.1% PBST) 1: 5000, an anti-hDHFR mouse antibody (sc-377091, Santa Cruz) diluted 1:5000 in 2.5% milk in PBST, anti-HSP70 mouse(AS08 371, Agrisera, generous gift of Goldberg lab) 1:1000 in 2.5% milk in PBST, anti-PfHAD1 rabbit antibody (generous gift of Odom lab) 1:10000 in 2.5% milk PBST. Secondary HRP-labeled anti-mouse or anti-rabbit antibodies are diluted 1:5000 in

2.5% milk in PBST and incubated for an hour. After incubation with the primary antibody, the PVDF membranes were washed three times for 5 min in PBST, Prepare Working Solution by mixing equal parts of the Stable Peroxide Solution and the Luminol/Enhancer Solution (34577 SUPERSIGNAL WEST PICO PLUS, 34096 SUPERSIGNAL WEST FEMTO MAXIMUM SENSITIVITY SUBSTRATE respectively). We incubate the blot in Working Solution for 5 min. Remove the blot from Working Solution and drain excess reagent. Afterward we tookimages were generated by BioRad Molecular Imager CHemiDoc XRS System with Image Lab software.

## Starvation assay

Asynchronous *P. falciparum* Dd2 parasites clones, with HA-tagged Pelota gene cultured in complete RPMI at 5% hematocrit and grown to ~3–5% parasitemia. The parasites were washed twice in PBS, equally partitioned, and washed in complete, isoleucine-free, and then were replated in their respective medium. Parasite cultures were incubated at 37°C, in a candle jar atmosphere for 48 hr. After incubation, parasites were harvested. After harvesting, infected RBCs were lysed for total RNA/protein isolation or polysome profiling (*Babbitt et al., 2012*).

## Polysome-associated RNA isolation

For polysome/RNA isolation we did according to published protocols (*Bunnik et al., 2013*; *Lacsina et al., 2011*). Shortly, cycloheximide (100 mM) was added to parasite-infected red blood cell cultures to a final concentration of 200 µM. The culture was incubated for 10 min at 37°C following with pelleting erythrocytes (5 min at 500 x g at room temperature) and washed twice in PBS containing 200 µM cycloheximide. After the last wash, pellets were kept on ice and were subsequently lysed by adding 2.2 volumes of lysis buffer (1% (v/v) Nonident P-40% and 0.5% (w/v) sodium deoxycholate in polysome buffer (400 mM potassium acetate, 25 mM potassium HEPES pH 7.2, 15 mM magnesium acetate, 200 µM cycloheximide, 1 mM dithiothreitol (DTT), and 1 mM 4-(2-aminoethyl) benzenesulfonyl fluoride HCl (AEBSF))) or cOmplete Protease Inhibitor Cocktail without EDTA (Roche), RNase OUT 1 µl/ml (Invitrogen). After 10 min incubation on ice, lysates were centrifuged for 15 min at 20,000xg at 4°C, at this point, the pellets were flash-freeze and stored at −80°C. The clarified lysates were then loaded on top of a sucrose cushion (35% sucrose in polysome buffer) to concentrate the ribosomes (4 ml polycarbonate ultracentrifuge tubes and then centrifuged for two h at 150000xg at 4°C in a Type 100.3 Ti rotor (Beckman Coulter, Brea, CA, USA). Ribosome pellets were resuspended in polysome buffer. Afterward, the ribosome suspension was layered on top of a 15 ml continuous linear 15% to 60% sucrose (w/v) 2 hr 40 min 260343xg (Beckman Optima XPN-90 and the swinging bucket rotor SW41 Ti). Fractions of 500 µl were collected using a UA-5 UV detector and model 185 gradient fractionator (ISCO, Lincoln, NE, USA). RNA was extracted with acid-phenol: chloroform pH 4.5 (Life Technologies), extracted twice with chloroform, and then precipitated using isopropanol.

## *Plasmodium falciparum* sucrose cushion for polysome profiling

This prep is used to isolate crude ribosome pellet, removing hemoglobin, which can then be later used for polysome profiling. We used 2 ml 100% hematocrit erythrocytes 8% parasitemia. The pellets were washed with 10 ml PBS and flash freeze in liquid nitrogen. Samples were stored at −80C before further use. Reagents: To make 10 ml *Plasmodium falciparum* Polysome Lysis Buffer (2.2 V/ sample), we used 25 mM K-HEPES (1 M stock) 250 µL, 400 mM K-OAc (4 M stock) 1000 µL, 15 mM Mg-OAc (1 M stock) 150 µL, 1% Igepal CA-360 (100% stock), 100 µL 0.5% Na Deoxycholate (10% stock) 500 µL, 200 µM cycloheximide (100 mM stock) 20 µL, 1 mM AEBSF (200 mM stock) 50 µL, 1 mM DTT (1 M stock) 10 µL, RNase Inhibitory (40 U/µL stock) 10 µL, Molecular Grade Water 7 mL, 910 µL. *Plasmodium falciparum* Sucrose Cushion for Polysome Profiling (15 ml), 25 mM K-HEPES (1 M stock) 375 µL, 400 mM K-OAc (4 M stock) 1500 µL, 15 mM Mg-OAc (1 M stock) 225 µL, 200 µM cycloheximide (100 mM stock) 30 µL, 1 mM AEBSF (200 mM stock) 75 µL, 1 mM DTT (1 M stock) 15 µL, 40 U/mL RNase Inhibitory (40 U/µL stock) 15 µL, Ultrapure sucrose 5.135 g, Molecular Grade Water (to start, complete to 15 mL after dissolved and all components added) 7.5 mL. *Plasmodium falciparum* Polysome Wash Buffer (1.5 mL/sample) 25 mL. 25 mM K-HEPES (1 M stock) 625 µL, 400 mM K-OAc (4 M stock) 2500 µL, 15 mM Mg-OAc (1 M stock) 375 µL, 200 µM cycloheximide (100 mM stock) 50 µL, 0.1 mM AEBSF (200 mM stock) 12.5 µL, 1 mM DTT (1 M stock) 25 µL, 10 U/mL

RNase Inhibitory (40 U/μL stock) 6.25 μL, Molecular Grade Water 21 mL. 406.25 μL *Plasmodium falciparum* Polysome Lysis Buffer (500 μL/sample) 2.5 mL 25 mM K-HEPES (1 M stock) 62.5 μL 400 mM K-OAc (4 M stock) 250 μL, *Plasmodium falciparum* Sucrose Cushion for Polysome Profiling - 15 mM Mg-OAc (1 M stock) 37.5 μL, 1% Igepal CA-360 (100% stock) 25 μL, 200 μM cycloheximide (100 mM stock) 5 μL, 1 mM AEBSF (200 mM stock) 12.5 μL, 1 mM DTT (1 M stock) 2.5 μL, 40 U/mL RNase Inhibitory (40 U/μL stock) 2.5 μL, Molecular Grade Water 2 mL and 102.5 μL. All steps were performed on ice. Pellets, 2 ml, were lysed with 2.2 V of lysis buffer, vortex to mix well. Incubate at 4 ˚C for 10 mins while rocking or rotating. Centrifuge 11,800 x g, move lysate to the fresh tube (~6.2 mL) For Sucrose Cushion Setup, we needed three cushions per sample. We were using 1 mL cushion per ~2 mL lysate. Add 1 mL sucrose to each tube. Layer sample lysates over sucrose cushion. Add 2 mL to each, then split the remainder (usually a few hundred microliters) over the three tubes. We used TLA 100.3 rotor (kept cold in the refrigerator) for ultracentrifugation. Speed: 100,000 x g, Time: 1 hr 30 mins 15 c. Temp: 4 ˚C When the centrifugation is finished, carefully aspirate off supernatant without touching sides so as not to disrupt pellets. Wash pellet 3X with 500 μL wash buffer, gently pipetting on the side opposite the pellet near the bottom of the tube. Resuspend and combine pellets for each sample in a total of 500 μL ribosome resuspension buffer. Pipet to disperse pellet. Use 200 μL tips to further disperse. Move to 1.5 mL microcentrifuge tube Incubate at 4 ˚C, rotating end-over-end, for at least 10 min. To remove remaining hemoglobin, set up another 1 mL sucrose cushion for each sample as previously. 23. A quick spin to collect samples to the bottom of the tube. Layer ribosome suspension over sucrose cushion, balance using ribosome resuspension buffer. Centrifuge as previously. Wash 3X with 500 μL wash buffer as previously. Resuspend in 500 μL ribosome resuspension buffer as previously. If not, proceed to the next steps immediately, flash-freeze in liquid nitrogen and store at −80 ˚C.

## Parasite live imaging

To image *P. falciparum* Dd2 strain episomally expressing polyA- and polyA+ constructs, we used 50 μl of infected erythrocytes, washed two times in PBS. The nucleus was stained with 1:1000 dilution of Hoechst 33342 for 10 min at room temperature. The cells were washed two times with PBS (500xg, 5 min, room temperature). After the washing step, the cells were resuspended in PBS (500 μl), 5 μl of cell resuspension was put on positively charged slides and put Zeiss cover glasses (item number:474030-9000-000). The cover glasses were sealed with nail polish, and subsequently, microscopy was performed.

Samples were visualized using an upright Zeiss Examiner.Z1-based 880 LSM with a 100x/1.46 oil-immersion objective. DAPI was excited using a 405 nm diode laser, mCherry was excited with a 561 nm DPSS laser, and GFP was excited with an Argon laser tuned to 488 nm. Optical sections (0.3 μm) were acquired with an Airyscan super-resolution detector and were processed using ZEN Blue v. 2.3.

## Generating *Tetrahymena thermophila* expressing YFP plasmids without (polyA-) and with (polyA+) coding for 12 lysines

*Tetrahymena thermophila* strain B2086 (II) was used for all experiments reported. Similar results were obtained with strain CU428 [(VII) mpr1-1/mpr1-1]. To assess the effect of LysAAA codons on protein accumulation, we modified a fluorescent protein tagging vector, pBSICY-gtw (*Motl and Chalker, 2011*) so as to fuse YFP to the carboxyl-terminus of a macronucleus-localized protein of unknown function (TTHERM_00384860), separated by a Gateway recombination cassette (Invitrogen/Life Technologies, Inc), and expressed from the cadmium inducible *MTT1* promoter (*Shang et al., 2002*). The TTHERM_00384860 coding region was amplified with oligonucleotides 5' ALM Bsi' 5' - CAC CCG TAC GAA TAA AAT GAG CAT TAA TAA AGA AGA AGT-3' and 3' ALM RV 5'- GAT ATC TTC AAT TTT AAT TTT TCT TCG AAG TTG C 3' and cloned into pENTR-D in a topoisomerase mediated reaction prior to digesting with BsiWI and EcoRV and inserting into BsiWI/PmeI digested pBSICY-gtw. Subsequently, LR Clonase II was used to insert a linker containing the sequence coding for an HA epitope tag alone (N) or the tag plus 36 adenosines (K12) in place of the Gateway cassette. The expression cassette is located within the 5' flanking region of a cycloheximide resistant allele of the rpL29 gene to direct its integration into this genomic locus. These constructs were linearized with PvuI and SacI in the region flanking the Tetrahymena rpl29 sequences and introduced into starved *Tetrahymena* cells by biolistic transformation (*Bruns and Cassidy-Hanley, 2000*;

*Cassidy-Hanley, 1997*). Transformants were selected in 1x SPP medium containing 12.5 µg/ml cyclo-heximide. To control for copy number, PCR assays with primers MTT2386 5'- tc tta gct acg tga ttc acg −3'and Chx-117, 5'- ATG TGT TAT TAA TCG ATT GAT −3' and Chx85r, 5'- TCT CTT TCA TGC ATG CTA GC – 3' verified that all rpL29 loci contained the integrated expression construct. Transgene expression was induced by addition of 0.4 µg/ml CdCl$_2$ and cells were grown 12–16 hr before monitoring protein accumulation. YFP accumulation was visualized by epifluorescence microscopy as previously described (*Matsuda et al., 2010*). Whole cells extracts were generated by boiling concentrated cell pellets in 1x Laemmli sample buffer, followed by were fractionation on 10% SDS polyacrylamide gels and transferred to nitrocellulose. YFP accumulation was a monitored with mouse anti-GFP antisera (G28R anti-GFP (OAEA00007) antibody, Aviva Systems Biology) and normalized to acetylated Rabbit anti-Histone H3 trimethyl-lysine (Upstate Biotechnologies/Millipore, NY, 07–473). Accumulation of transcripts was assessed on Northern blots as previously described using a $^{32}$P-random-primer-labeled YFP probe (*Chalker and Yao, 2001*).

## 2D-electrophoresis

2 ml (100% hct) of *P. falciparum* infected erytrocyes (HA-pelo Dd2 strain, parasitemia 7–9%) were treated with PBS saponin 0.15%, and protease inhibitor w/o EDTA. The pellets were washed in PBS-saponin bufer three times. Afterwards, the samples were used to do immunoprecipitation (IP). We used Pierce HA-magnetic beads 25 µl and further proceeded with cell lysis:Lysis Buffer:150 mM NaCl, 50 mM Tris pH 7.5, 1% IGPAL-CA-630 (Sigma, #I8896), 5% Glycerol, Protease (1 mM PMSF) and phosphatase inhibitors. Sample wash: Wash Buffer: 150 mM NaCl, 50 mM Tris pH 7.5, 5% Glycerol. After the washing step the samples were snap frozen in liquid nitrogen for 2D gel analysis.

Two-dimensional electrophoresis was performed according to the carrier ampholine method of isoelectric focusing (*O'Farrell, 1975*; *Burgess-Cassler et al., 1989*) by Kendrick Labs, Inc (Madison, WI) as follows: Isoelectric focusing was carried out in a glass tube of inner diameter 2.0 mm using 2% pH 3–10 isodalt servalytes (Serva, Heidelberg, Germany) for 9600 volt-hrs. One µg of an IEF internal standard, tropomyosin, was added to the sample. This protein migrates as a doublet with lower polypeptide spot of MW 33,000 and pI 5.2. The enclosed tube gel pH gradient plot for this set of ampholines was determined with a surface pH electrode.

After equilibration for 10 min in Buffer 'O' (10% glycerol, 50 mM dithiothreitol, 2.3% SDS and 0.0625 M tris, pH 6.8), each tube gel was sealed to the top of a stacking gel that overlaid a 10% acrylamide slab gel (0.75 mm thick). SDS slab gel electrophoresis was carried out for about 4 hr at 15 mA/gel. After slab gel electrophoresis, the duplicate gel for blotting was placed in transfer buffer (10mMCaps, pH 11.0, 10% MeOH) and transblotted onto a PVDF membrane overnight at 200 mA and approximately 100 volts/two gels. The following proteins (Millipore Sigma) were used as molecular weight standards: myosin (220,000), phosphorylase A (94,000), catalase (60,000), actin (43,000) carbonic anhydrase (29,000) and lysozyme (14,000). These standards appear as bands at the basic edge of the Coomassie Brilliant Blue R-250-stained membrane.

## Coding sequence metrics

Coding sequences for each organism (*Plasmodium falciparum* 3D7 - EPr 1, *Tetrahymena thermophila* – JCVI-TTA1-2.2, and human - GRCh38.p7) were downloaded from their respective Ensembl BioMart pages (accessed on March 19[th] 2018). Each set of genes was filtered to include only the longest coding sequence variant. The resulting sequences were each analyzed using a folding window approach; a window of 120 nt was scanned across each sequence with a single nucleotide step size. Each window sequence was folded (using RNAfold) to determine its minimum free energy structure/value (MFE$_{native}$) (*Lorenz et al., 2011*). At the same time, MFE values were calculated for 30 randomized versions of the native window sequence (MFE$_{random}$). The mean of MFE$_{random}$ values were compared to that of the native in a method adapted from *Clote et al., 2005* and illustrated in the following equation:

$$z - score = \frac{MFE_{native} - \overline{MFE_{random}}}{Std\ Dev_{MFE}}$$

The mean window MFE and z-score values of each gene were calculated and compiled in

separate spreadsheets. Box and whisker plots (generated using BoxPlotR *Spitzer et al., 2014*) are shown, depicting the distribution of these means for each organism.

## Bioinformatics analyses

### Ribosome profiling

GWIPS database was used as the source of ribosome profiling data. We downloaded *Caro et al., 2014* dataset for *Plasmodium falciparum* (the only dataset available for that species), while for humans, we used aggregate for all deposited studies. In both cases, we took a dataset for elongating ribosomes mapped to A-site.

The definition of polyA-tracts was taken from *Arthur et al., 2015*, which are twelve consecutive adenines allowing for one mismatch. Genomic coordinates of such segments were downloaded from PATACSDB (*Habich et al., 2016*). We took 50 residues upstream and downstream from the beginning of the polyA segment, preserving the proper strand orientation. Occupancy plot was generated with two conditions:

1. discarding fragments that had less than five occupancy values in the given window of 101 nucleotides
2. taking into account fragments that had the average occupancy equal or higher than the mean for the dataset (this was to remove the influence of poorly mapped segments from the plots)

To make occupancy plots between different life stages comparable (*Figure 4—figure supplement 2*), we have introduced a normalization mechanism, where occupancy of polyA region was divided by a mean from similarly sized distribution of random fragments of the same length that had the 0 position within the coding region. Randomization was preserving chromosome distribution of the original polyA-carrying genes dataset. All the analyses were done using R language.

### Gene ontology analysis

Gene Ontology analysis was done using the Gene Ontology Enrichment tool at PlasmoDB website (*Aurrecoechea et al., 2009*), using default options. For *Supplementary file 1* we took only those terms that had Bonferroni-adjusted p-value better than 0.05.

### Structural analyses

Structural analyses were done using MOLEonline service version 2.5 (*Pravda et al., 2018*) using default parameters, with manual selection of the proper tunnel among all predicted by this service. Due to limits of the method, it was not possible to enforce the same length of the tunnel in all structures, yet the hydrophobic patches formed by interaction with rRNA and L4/L22 proteins were easy to observe in all structures.

## Acknowledgements

We thank Hani Zaher as well as members of Daniel Goldberg's and Sergej Djuranovic's lab for helpful comments. This work is supported by NIH R01 GM112824 to SD, NIH R00 R00GM112877 to WM, NSF MCB 1412336 to DLC, and NIH T32 GM: 007067 to JE. POB and JAJF are supported by the Washington University Center for Cellular Imaging (WUCCI), which is funded in part by Washington University School of Medicine, The Children's Discovery Institute of Washington University, and St. Louis Children's Hospital (CDI-CORE-2015–505 and CDI-CORE-2019–813) and the Foundation for Barnes-Jewish Hospital (3770). The authors declare that they have no competing interests.

## Additional information

### Funding

| Funder | Grant reference number | Author |
| --- | --- | --- |
| National Institute of General Medical Sciences | R01 GM112824 | Sergej Djuranovic |
| National Science Foundation | MCB 1412336 | Douglas L Chalker |

| National Institute of General Medical Sciences | R00 GM112877 | Walter N Moss |
| National Institute of General Medical Sciences | T32 GM007067 | Jessey Erath |
| St. Louis Children's Hospital | CDI-CORE-2015-505 | Peter O Bayguinov James AJ Fitzpatrick |
| St. Louis Children's Hospital | CDI-CORE-2019-813 | Peter O Bayguinov James AJ Fitzpatrick |
| Washington University School of Medicine in St. Louis | | Peter O Bayguinov James AJ Fitzpatrick |
| Foundation for Barnes-Jewish Hospital | 3770 | Peter O Bayguinov James AJ Fitzpatrick |

The funders had no role in study design, data collection and interpretation, or the decision to submit the work for publication.

## Author contributions

Slavica Pavlovic Djuranovic, Conceptualization, Data curation, Formal analysis, Supervision, Validation, Investigation, Visualization, Methodology, Writing - original draft, Writing - review and editing; Jessey Erath, Conceptualization, Data curation, Formal analysis, Validation, Investigation, Visualization, Methodology, Writing - original draft, Writing - review and editing; Ryan J Andrews, Data curation, Software, Formal analysis, Supervision, Funding acquisition, Validation, Investigation, Visualization, Writing - original draft, Writing - review and editing; Peter O Bayguinov, Formal analysis, Investigation, Visualization, Writing - review and editing; Joyce J Chung, Formal analysis, Investigation, Writing - original draft; Douglas L Chalker, Formal analysis, Validation, Investigation, Writing - original draft, Writing - review and editing; James AJ Fitzpatrick, Formal analysis, Funding acquisition, Validation, Investigation, Visualization, Writing - review and editing; Walter N Moss, Data curation, Formal analysis, Investigation, Visualization, Writing - original draft, Writing - review and editing; Pawel Szczesny, Data curation, Software, Formal analysis, Validation, Investigation, Visualization, Methodology, Writing - original draft, Writing - review and editing; Sergej Djuranovic, Conceptualization, Data curation, Formal analysis, Supervision, Funding acquisition, Validation, Investigation, Visualization, Writing - original draft, Project administration, Writing - review and editing

## Author ORCIDs

Jessey Erath ![ORCID] https://orcid.org/0000-0001-8802-635X
Ryan J Andrews ![ORCID] http://orcid.org/0000-0003-0275-0019
Douglas L Chalker ![ORCID] http://orcid.org/0000-0002-0285-3344
Walter N Moss ![ORCID] http://orcid.org/0000-0001-6419-5570
Sergej Djuranovic ![ORCID] https://orcid.org/0000-0002-9417-0822

## Decision letter and Author response

Decision letter https://doi.org/10.7554/eLife.57799.sa1
Author response https://doi.org/10.7554/eLife.57799.sa2

## Additional files

### Supplementary files

• Supplementary file 1. Table indicating highly significant gene ontology terms (GO) from biological process category for polyA tract carrying genes in *Plasmodium falciparum*. Background counts (Bgd count) represent all genes in defined GO group, result count are all genes with polyA tract found in identified GO group, percent of background (Pct of bgd) represents percent of polyA track genes in GO group. Fold enrichment is calculated over all polyA track genes as well as odd ratio. P-value as well as Benjamini and Bonferroni test values indicate statistical significance of GO analyses.

• Supplementary file 2. Comparison of factors associated with human NGD/NSD and RQC pathways and bioinformatics in *Plasmodium spp*-associated PlasmoDB database. *P. falciparum* genes related to human counterparts are given in brackets.

• Transparent reporting form

### Data availability

All data generated or analysed during this study are included in the manuscript, supporting files or referenced. Source data files have been referenced for Figures 1, 3 and 5, as well as for figure supplements.

The following previously published datasets were used:

| Author(s) | Year | Dataset title | Dataset URL | Database and Identifier |
|---|---|---|---|---|
| Caro F, Ahyong V, Betegon M, DeRisi JL | 2014 | Ribosome Profiling in P. falciparum asexual blood stages | https://www.ncbi.nlm.nih.gov/geo/query/acc.cgi?acc=GSE58402 | NCBI Gene Expression Omnibus, GSE58402 |

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
