## [Decision Letter]

**Acceptance summary:**

*Plasmodium falciparum* genome is extremely AT-rich (81% of the genome). PolyA tracts in open reading frames in eukaryotes and prokaryotes trigger ribosome quality control mechanisms. The authors show that in contrast to other organisms, in *Plasmodium falciparum*, mRNAs containing long polyA sequences are efficiently and accurately translated. Also, polyA tracts fail to engender No Go Decay (NGD) process or production of frameshift-protein fragments.

**Decision letter after peer review:**

[Editors’ note: the authors submitted for reconsideration following the decision after peer review. What follows is the decision letter after the first round of review.]

Thank you for submitting your work entitled "*Plasmodium falciparum* translational machinery condones polyadenosine repeats" for consideration by *eLife*. Your article has been reviewed by two peer reviewers, and the evaluation has been overseen by a Reviewing Editor and a Senior Editor. The reviewers have opted to remain anonymous.

Our decision has been reached after consultation between the reviewers. Based on these discussions and the individual reviews below, we regret to inform you that your work cannot be considered further for publication in *eLife*, at least in its current form.

In general, the reviewers agree that the main observation of the absence of mRNA decay of RQC substrates in *Plasmodium* is very interesting and that the field would greatly benefit from knowing it. However, in the present form, the ribosome profiling data are unreliable due to the pre-treatment with cycloheximide. The analysis in Figures 3-4 is significantly lacking. Evidence should be obtained to determine whether ribosomes are actually stalling or not. The *eLife* rules require resubmission of a revised version of the paper in 2 months. This would appear to be hard to accomplish. We hope that you can respond to the reviewers' criticisms given extra time. As the reviewers' comments are extensive but highly constructive, they are detailed below. A revised version of your manuscript that addresses all these issues would likely be suitable for reconsideration.

Reviewer #1:

Ribosome stalling and the RQC pathway, which resolves the stalled ribosome and degrade the mRNA and nascent peptide, have drawn considerable attention in recent years. However, there are significant gaps in our understanding of the various aspect of these phenomena, particularly their importance in physiology, considering that their substrate mRNAs with runs of multiple AAA codons, are quite rare in eukaryotic genomes. Therefore, it has been previously proposed that RQC on mRNAs with polyA sequence within ORFs may have anti-parasitic significance. In this manuscript, Djuranovic and colleagues demonstrate a very interesting observation that the malaria-causing parasite *P. falciparum*, circumvent the ribosome stalling process at polyA sites, despite having a remarkable number of mRNAs with long polyA tracks. Thus, this manuscript clearly advances our understanding of this process and extend the field further and should be of interest to investigator in various fields. However, despite its strengths in providing convincing data on the unusual nature of the *P. falciparum* transcriptome and the apparent deficiency of the mRNA decay pathway in this species, the manuscripts does not properly address the other possible aspect of the RQC-induced silencing; mRNA translation. In addition, the robustness of the data produced by ribosome profiling assay maybe compromised due to technical issues as described below. Therefore, the manuscript may benefit from additional experimentation/analyses to verify this aspect.

1) The contribution of translational repression in this process is not properly addressed. This is an important issue in light of the emphasize that the authors placed on the main role of the mRNA sequence, rather than amino acid sequence of the nascent peptide, in stalling. Nevertheless, while the authors quite clearly demonstrate the lack of efficient mRNA decay pathway, their explanation for the absence of stalling at polyA sequences in *P. falciparum* seems to rely solely on the differences in the exit channel of the ribosome. This comes across as changing the goal post. Rather than explaining why the mRNA sequence does not induce stalling, the authors in fact try to address why the peptide doesn't induce stalling. On this note, it is unclear why the authors focused only on measuring the mRNA levels of a few of the candidate genes with more than 22As (Figure 4). It is quite possible that due to deficiency in the mRNA decay pathway, the target mRNAs are only translationally repressed, but it does not rule out the potential translational repression. As it stands, it is unclear whether the ribosome stalling and RQC pathway in *P. falciparum* is deficient, or the observed phenomenon is merely due to the deficient mRNA decay. Therefore, performing a polysome profiling assay and western blot for the candidate proteins would be suitable additions to confirm the lack of stalling on mRNAs with long polyA tracts in *P. falciparum*.

2) It has been previously shown that pre-treatment of the samples with translation inhibitors such as cycloheximide and anisomycin could complicate the analysis of ribosome stalling by ribosome profiling assay and result in false negative outcome, when investigating ribosome pausing/stalling on poly-basic sites (Requião, et al., 2016). From what could be deduced in the Materials and methods section, the samples seem to have been pre-treated with cycloheximide for 10 minutes before collection. Therefore, this may in fact take away from the reliability of the observation that there is no ribosome pausing at polA tracts in *P. falciparum* based on the ribosome profiling assay, as it may be a false negative observation due to artefacts caused by cycloheximide.

Reviewer #2:

In this manuscript, Pavlovic-Djuranovic et al. show that poly(A) tracks in *Plasmodium falciparum* are efficiently translated and their translation does not trigger No Go Decay (NGD). This is very interesting since translation of poly(A) sequence elicits NGD in human cell or in *Tetrahymenathermophila*, which is an organism with an AT-content similar to *P. falciparum*. The authors further show that isoleucine starvation causes upregulation of Pelo but suggest this Pelo is not involved with NGD. Inserting a stem loop in their reporter construct disrupts protein synthesis but still does not trigger NGD. The authors hypothesize that the ribosome rescue pathway is evolved in this parasite differently than other organisms and NGD is not triggered upon ribosome stalling. They speculate that loss of RACK1's association with the ribosome and other structural differences could underlie these trends, though no data are offered to test these hypotheses. Overall, this paper grapples with an intriguing observation and improves our understanding of ribosome quality control system in different organisms. However, the paper seems to have been quickly written and lacks clarity. While much of the data are negative, it is worth sharing if further experiments could be performed in some cases to more clearly explain what models are being tested and directly show the results in *Plasmodium* are unlike those in other organisms.

List of major concerns:

Is it possible that during poly(A) translation, *P. falciparum* ribosomes miscode and incorporates amino acids other than Lys? A mass spectrometry analysis of the reporter peptide could address this possibility and show that the ribosome faithfully translates Lys codons without miscoding and frameshifting.

It is difficult to see the difference between human and *P. falciparum* ribosome occupancy in Figures 3A and 3B. It should be shown by distinct colors and the color code should be included in the figure legends. What do the error bars signify? Is the large central dip in 3B significant? Also, the authors should explain how many genes are going into the analysis and how they are controlling for the possibility that a few strongly-expressed genes are dominating these effects (how was normalization done?).

The authors refer to previous studies where frameshifting was attributed to translation of poly(A) sequences. They do not see frameshifting when poly(A) reporters were examined here in *Plasmodium*. However, they do not show that their reporters do cause frameshifting in human cells. Is there a positive control to show these reporters can cause it? (Figure 2, Figure 2—figure supplement 2, Figure 2—figure supplement 3). If not, perhaps the language should be adjusted to not limit the effects to frameshifting.

The authors do not show or cite evidence that Ile-starvation leads to substantial ribosome stalling in their cells. They also do not cite cases where Pelo has been observed to target ribosomes that are stalled awaiting an aa-tRNA. Gcn1 and Gcn20 have been thought to directly sense stalled ribosomes under these conditions. Are those factors present here? Is eIF2α phosphorylated and could Pelo be downstream of ATF4?

The argument that overexpression of Pelo can stimulate mRNA degradation has not been previously shown (i.e. a study of mRNA stability in an overexpression strain). The DHH8 mRNA level could be increased during Ile starvation for any number of reasons and the authors have not shown ribosome stalling on this transcript is occurring. The authors should provide results from other Ile codon containing transcripts or correlate mRNA levels with Ile content globally by RNA-seq. It's not clear this is the best test for NGD.

Figure 4C: the stem loop construct, HA-pelo protein level seems to be increased – why is this occurring? Is this the natural variation between experiments? If so, how do we know the difference in 4B is significant? Were replicates done? Also, it is concerning that +polyA_36_ containing peptide migrates to the same exact place in the gel as -polyA_36_ peptide despite additional 12 positively charged amino acids. Is this just a resolution issue in this particular gel? Also, what is the band migrating at ~15 kD and why could it be changing with different constructs? Finally, there is a faint band that is just a bit heavier than the HA-mCherry product – is this the reporter with stem loop present?

The section on RNA structure and, particularly, z-scores for transcript stability was not very comprehensible. It appears the authors were asking whether poly(A) regions ("native" regions?) were more or less stable than randomly selected regions, but I could be wrong. Overall, this section did not seem to make any compelling arguments toward explaining the lack of no-go decay in this organism.

[Editors’ note: further revisions were suggested prior to acceptance, as described below.]

Thank you for submitting your article "*Plasmodium falciparum* translational machinery condones polyadenosine repeats" for consideration by *eLife*. Your article has been reviewed by two peer reviewers, and the evaluation has been overseen by a Reviewing Editor and James Manley as the Senior Editor The reviewers have opted to remain anonymous.

The reviewers have discussed the reviews with one another and the Reviewing Editor has drafted this decision to help you prepare a revised submission.

Revisions:

The two original reviewers recommend publication of your new version of the manuscript based on the additional conclusive evidence that you provided. The second reviewer provides comments on the text and offers excellent suggestions for improvements, vis-a-vis the explanation for the rationale for experiments. Also, some of the later figure panels might be better added to supplemental materials. The comments are noted below. We hope you find these suggestions helpful.

Reviewer #1:

The authors have mainly answered my major concerns in the first version of the manuscript, particularly regarding the potential artifacts caused by cycloheximide pretreatment.

Reviewer #2:

The revised manuscript addressed the major concerns by performing new experiments (polysome profiling, 2D gels) and by including additional controls. The text and figures are improved, and the story is now more compelling. The remaining issues are fixable and the most significant one is the writing, which still does not offer clear rationales for why some experiments were performed and tends to overinterpret results. In addition, the writing suffers from many typographic errors (some noted below) and some figures can benefit from a better presentation.

1) Figure 4: While the explanation of methodology is improved, I'm not sure what information Figures 4C and 4D add to the story. The text implies that the AT-rich genome could affect structure of mRNAs (less stable). The rationale for this analysis was not clearly stated. It is implied that if evolution has compensated for structure, there would be an effect in the z-scores. But how would this be related to assays of stalling? What is the predicted observation? Since the assay is indirect (structure is not measured) and the purpose is not clearly articulated, I suggest either placing these figures into supplementary or removing them from the manuscript.

2) Figure 6: The authors state that previous microarray analysis in *Plasmodium* showed that Ile starvation does not globally change mRNA levels. Therefore, the rationale to choose this experiment to study NGD in *P. falciparum* is unclear. Why did the authors expect changes on Ile-rich mRNAs given the prior study already showed "similar overall levels for the majority of mRNA transcripts"? Second, is it known that in mammalian cells that aa starvation induces NGD? If so, that should be clearly cited and explained. Alternatively, amino acid starvation could allow uncharged tRNAs to fill the A-site and block Pelota binding, so positive control data in mammalian cells establishing this expectation would add confidence. Third, in yeast, the mRNA stabilizing effects of loss of Dom34 is not that strong and I'm not sure anyone has observed increased mRNA degradation in a Dom34/Pelota overexpression experiment. If such data exists, it should be cited, as it would add confidence. Without offering a stronger rationale for these experiments, the conclusions from this figure are highly speculative and the authors should soften the claims and possibly move it to supplementary data.

3) It is widely accepted that NGD is part of RQC and the authors should clarify that, as previously requested, in the Introduction.

4) To make Figures 4A and 4B clearer, I recommend directly adding labels for *H. sapiens* and *P. falciparum* on top of the graphs and improving the figure legends. It's not clear that the first sentence of the legend refers to both A and B. It's not clear what "taken into account" means without context of the main text. Showing full, uncut error bar in Figure 4B and then providing an inset with the y-axis occupancy of 200 may help readers to better grasp this figure since the boxes extending off the plot is misleading. Also, the technical artifact leading to the dip should be acknowledged in the legend. It's not clear why this artifact is not also present in A and therefore whether this artifact is masking the pause in B. To make this more convincing, the authors could create a control dataset where pauses were computationally introduced and then successfully detected (i.e. a positive control). Alternatively, some explanation should be offered in the text as to why this dip does not void this analysis.

---

## [Author Response]

[Editors’ note: the authors resubmitted a revised version of the paper for consideration. What follows is the authors’ response to the first round of review.]

In general, the reviewers agree that the main observation of the absence of mRNA decay of RQC substrates in Plasmodium is very interesting and that the field would greatly benefit from knowing it. However, in the present form, the ribosome profiling data are unreliable due to the pre-treatment with cycloheximide. The analysis in Figures3-4 is significantly lacking. Evidence should be obtained to determine whether ribosomes are actually stalling or not.

In our revised manuscript, we did multiple experiments to address reviewer's comments. Two major arguments of reviewer’s were, lacking of evidence for ribosomal stalling and the use of cycloheximide in the ribosome profiling dataset from *Plasmodium falciparum* (previously published in Caro et al., 2014). We should point out that there are articles that indicate that cycloheximide artifacts (described in Requião et al., 2016 or Santos et al., NAR, 2019) might be restricted to *S. cerevisiae* cells, ie: Sharma et al., bioRxiv 746255; doi: https://doi.org/10.1101/746255; Duncan and Mata, Sci. Rep., 2017; https://doi.org/10.1038/s41598-017-10650-1.

As seen from the above articles cycloheximide artifacts are not present in the ribosome profiling data from humans and other species and are somewhat challenging to infer to *P. falciparum.* However, we have decided to address this problem experimentally. We have now included experiments to show that *P. falciparum* ribosomes can be stalled without inducing targeted mRNA degradation. We did polysome profiles without pre-incubation of parasites with cycloheximide and inspected the distribution of wild type (-polyA_36_), polyA tract (+polyA_36_) and stem loop (StL) transcripts. We found that stem loop mRNA distribution correlates mostly with disomes, while polyA and will type constructs are found in higher polysome fractions. These results are now included in Figure 5. These results strengthens our data that polyA tracts are not stalling *Plasmodium* ribosomes. The data on stem-loop construct further indicate that the stalling of ribosomes, and reduced protein production, in *Plasmodium* cells does not lead to targeted mRNA degradation (as shown in our original data). This answers also general comment to our original submission, whether the P*lasmodium* ribosome are able to stall or not.

Reviewer #1:[…]1) The contribution of translational repression in this process is not properly addressed. This is an important issue in light of the emphasize that the authors placed on the main role of the mRNA sequence, rather than amino acid sequence of the nascent peptide, in stalling. Nevertheless, while the authors quite clearly demonstrate the lack of efficient mRNA decay pathway, their explanation for the absence of stalling at polyA sequences in *P. falciparum* seems to rely solely on the differences in the exit channel of the ribosome. This comes across as changing the goal post. Rather than explaining why the mRNA sequence does not induce stalling, the authors in fact try to address why the peptide doesn't induce stalling. On this note, it is unclear why the authors focused only on measuring the mRNA levels of a few of the candidate genes with more than 22As (Figure 4). It is quite possible that due to deficiency in the mRNA decay pathway, the target mRNAs are only translationally repressed, but it does not rule out the potential translational repression. As it stands, it is unclear whether the ribosome stalling and RQC pathway in *P. falciparum* is deficient, or the observed phenomenon is merely due to the deficient mRNA decay. Therefore, performing a polysome profiling assay and western blot for the candidate proteins would be suitable additions to confirm the lack of stalling on mRNAs with long polyA tracts in *P. falciparum*.2) It has been previously shown that pre-treatment of the samples with translation inhibitors such as cycloheximide and anisomycin could complicate the analysis of ribosome stalling by ribosome profiling assay and result in false negative outcome, when investigating ribosome pausing/stalling on poly-basic sites (Requião, et al., 2016). From what could be deduced in the Materials and methods section, the samples seem to have been pre-treated with cycloheximide for 10 minutes before collection. Therefore, this may in fact take away from the reliability of the observation that there is no ribosome pausing at polA tracts in *P. falciparum* based on the ribosome profiling assay, as it may be a false negative observation due to artefacts caused by cycloheximide.

In response to reviewer #1 major comments, we have now included polysome profiles without cycloheximide pre-treatment and look for the distribution of wild type (-polyA_36_), polyA (+polyA_36_) and stem-loop (StL) constructs. We followed the distribution of all transcripts, control transcript (GAPDH) and *Plasmodium* ribosomes by qRT-PCR. We found that stem loop construct distribution correlates mostly with disomes, while polyA and wild type constructs are found evenly distributed in polysomes. This strengthens our data that polyA tracts are not stalling *Plasmodium* ribosomes. This also further indicates that the stalling of ribosomes with stem loop in *Plasmodium* cells does not lead to targeted degradation (our original data). We are puzzled with reviewer’s comment:

“It is quite possible that due to deficiency in the mRNA decay pathway, the target mRNAs are only translationally repressed, but it does not rule out the potential translational repression”.

If the target polyA tract mRNAs are indeed translationally repressed, we should see much less protein being produced from our reporters (shown in Figure 2 and 3, and Figure 2—figure supplement 2 and 3), as well as in ribosome profiling data shown in Figure 4. The western blot analyses of reporters show opposite, protein synthesis from polyA tract transcripts is comparable, if not enhanced, compared to synthesis from transcripts without polyA tracts. The same is through in ribosome profiling data where polyA tract constructs from human tissue cultures indicate ribosome stalling (Figure 4A, regardless of cycloheximide treatment) while *P. falciparum* transcripts do not (Figure 4B). This analysis is plotted for 1167 transcripts from the *P. falciparum* cells with polyA tracts shorter than 22 adenosine nucleotides (22A-1 sequences). While we would be happy to test the expression of some of these endogenous constructs by western blot analyses, there are only a few available antibodies for *P. falciparum* proteins. PfHAD1 and PfHSP70 are used in our western blots, but unfortunately, these proteins do not host polyA tracts. However, our western blot analyses are done on reporters with and without polyA tract with 36 consecutive adenosines. We have commented on cycloheximide treatment in our general response.

Reviewer #2:[…] However, the paper seems to have been quickly written and lacks clarity. While much of the data are negative, it is worth sharing if further experiments could be performed in some cases to more clearly explain what models are being tested and directly show the results in Plasmodium are unlike those in other organisms.

We are thankful for reviewer #2 for comments on our manuscript. We have now added several additional experiments that further support our original data and clarify our major points: 1. Lack of ribosomal stalling on polyA tracts and poly-lysine repeats in *P. falciparum* cells; 2. Lack of targeted mRNA degradation of ribosome stalled transcripts. We have also increased the clarity of our manuscript by further changing original text.

List of major concerns:Is it possible that during poly(A) translation, *P. falciparum* ribosomes miscode and incorporates amino acids other than Lys? A mass spectrometry analysis of the reporter peptide could address this possibility and show that the ribosome faithfully translates Lys codons without miscoding and frameshifting.

In response to the reviewer’s comment, we have analyzed our reporter constructs using 2D gels to show whether fidelity and quality of protein synthesis stay intact in *Plasmodium* cells. We show now in the new Figure 3 that regardless of the lysine codons (AAA or AAG) both polylysine containing proteins shift their position from the wild-type (-polyA_36_) control by +1 overall charge in isoelectric point (pI for wild-type 6.2, pI for 12xLys 7.2). The position of protein products from 12xAAA and 12xAAG is identical on those gels arguing for no significant miscoding and frameshifting on either of these constructs. If such events do occur, they do not contribute significantly to overall quality or change in protein expression of polyA tract constructs.

It is difficult to see the difference between human and *P. falciparum* ribosome occupancy in Figures 3A and 3B. It should be shown by distinct colors and the color code should be included in the figure legends. What do the error bars signify? Is the large central dip in 3B significant? Also, the authors should explain how many genes are going into the analysis and how they are controlling for the possibility that a few strongly-expressed genes are dominating these effects (how was normalization done?).

Both figures are symmetrical, with the center of the plot being the beginning of the polyA segment, left side being upstream of polyA segment. Both plots intentionally use the same y-axis scale to enable comparison. Barplots at each position denote first and third quantile (box ranges), the median is denoted with the horizontal line inside a box. Vertical lines denote 1.5*IQR range from the box (standard box plot representation). There was no normalization done (other than applied by GWIPS pipeline), only removing of entries without occupancy or with low average occupancy, to improve signal to noise ratio (as described in the Materials and methods). For a human this resulted in the plot based on 32 polyA fragments (32 transcripts), out of 102, that had occupancy at the start if the polyA segment more significant than 0. This is due to the fact that polyA tract genes are generally not expressed well in most of the organisms and, as such, have reduced amount of reads over polyA segments in ribosome profiling data sets. *P. falciparum* analyses is done on 1167 genes. That is why outer whiskers (1.5*IQR lines) extend beyond the plot because there are many more data points with good occupancy. *Plasmodium* plot has a central dip due to the inability of bioinformatics software to correctly map very long polyA fragments (as shown in Figure 4—figure supplement 1). A similar procedure of removing poorly mapped fragments was applied here as well. There was no statistical assessment done in these plots – they simply show the difference between occupancy distributions in these two species.

The authors refer to previous studies where frameshifting was attributed to translation of poly(A) sequences. They do not see frameshifting when poly(A) reporters were examined here in Plasmodium. However, they do not show that their reporters do cause frameshifting in human cells. Is there a positive control to show these reporters can cause it? (Figure 2, Figure 2—figure supplement 2, Figure 2—figure supplement 3). If not, perhaps the language should be adjusted to not limit the effects to frameshifting.

Ribosomal frameshifting on polyA tracts was previously shown in *E. coli*, yeast, *Drosophila* and human cells with the identical or similar constructs, as well as endogenous human genes (ZCRB1). These experiments are described in Arthur et al., 2015; Koutmou et al., 2015; as well as Sundaramoorthy et al., 2017. We should also point out that ribosomal frameshifting was seen on sequences shorter than 36 consecutive adenosine constructs starting with 3 or 4 consecutive lysine AAA codons (9 or 12A polyA tracts).

The authors do not show or cite evidence that Ile-starvation leads to substantial ribosome stalling in their cells. They also do not cite cases where Pelo has been observed to target ribosomes that are stalled awaiting an aa-tRNA. Gcn1 and Gcn20 have been thought to directly sense stalled ribosomes under these conditions. Are those factors present here? Is eIF2α phosphorylated and could Pelo be downstream of ATF4?

Dan Goldberg’s lab originally reported Ile-induced starvation in *P. falciparum* (cited as Babbitt et al., 2012). This manuscript argues that Ile-starvation of *P. falciparum* cells is able to induce a state of hibernation through the arrest of protein synthesis with phosphorylation of eIF2α. This phosphorylation of eIF2α in *Plasmodium* is happening in a GCN2-independent fashion, and without existing TOR-nutrient sensing pathway nor activation of ATF4 homolog. The same manuscript gives micro-array analyses of transcript levels that also indicate no significant change in the mRNA levels of the majority of *P. falciparum* genes during the 48 hours starvation period (our focused qRT-PCR analysis in Figure 5C).

The argument that overexpression of Pelo can stimulate mRNA degradation has not been previously shown (i.e. a study of mRNA stability in an overexpression strain). The DHH8 mRNA level could be increased during Ile starvation for any number of reasons and the authors have not shown ribosome stalling on this transcript is occurring. The authors should provide results from other Ile codon containing transcripts or correlate mRNA levels with Ile content globally by RNA-seq. It's not clear this is the best test for NGD.

We have now included qRT-PCR analyses for additional eight *P. falciparum* genes with multiple runs of Ile-residues (3-7 consecutive Ile). As noted from the new Figure 5C, the general trend is rather no-change or stabilization of Ile-rich transcripts during the Ile-starvation experiment. We have also expanded our analyses of stem loop constructs and included new Figure 4C, which indicates the stalling of ribosomes on the stem loop transcript in *P. falciparum* cells.

Figure 4C: the stem loop construct, HA-pelo protein level seems to be increased – why is this occurring? Is this the natural variation between experiments? If so, how do we know the difference in 4B is significant? Were replicates done? Also, it is concerning that +polyA_36_ containing peptide migrates to the same exact place in the gel as -polyA_36_ peptide despite additional 12 positively charged amino acids. Is this just a resolution issue in this particular gel? Also, what is the band migrating at ~15 kD and why could it be changing with different constructs? Finally, there is a faint band that is just a bit heavier than the HA-mCherry product – is this the reporter with stem loop present?

We have not noticed an increase in the Pelo transcript or protein in parasites transfected with stem-loopp construct. HA-pelo was supposed to be normalization control in Figure 4C (now Figure 5A) and difference for Pelo protein is due to the differences in loading amounts of samples. The difference in HA-pelo amounts and transcript levels in Figure 6A and 6B (or previous Figure 4A and 4B) is a consequence of 48-hour Ile-starvation of parasites. As noted in figure legends, these experiments were done in 3 biological replicates. We think that no obvious change between -polyA_36_ and +polyA_36_ in this gel is due to the resolution issue. We have now also included 2D gels (Figure 3) of -polyA_36_, +polyA_36_ constructs and 12xAAG Lys construct to indicate a shift in isoelectric point of polylysine protein. 15kD band appearing in gels and causing cross-reactivity in western blots is contaminant of human hemoglobin. The *Plasmodium* parasite cultures are grown in human erythrocytes (98% of total protein is hemoglobin) and while we tried to lyse erythrocytes and get rid of most of hemoglobin prior to parasite lysis the amount of hemoglobin present in lysed samples is very hard to control. This is why we used PfHAD1 and endogenously tagged HA-pelo as normalization controls for western blot analyses. Finally, we would like to think that the faint band in Figure 5A is indeed HA-mCherry protein from stem-loop construct (StL). We just did not want to speculate whether this band indeed represents protein product from StL construct. We have also tried HA-immunoprecipitations with lysates expressing this particular construct (StL) without clear enrichment of protein corresponding to this band.

The section on RNA structure and, particularly, z-scores for transcript stability was not very comprehensible. It appears the authors were asking whether poly(A) regions ("native" regions?) were more or less stable than randomly selected regions, but I could be wrong. Overall, this section did not seem to make any compelling arguments toward explaining the lack of no-go decay in this organism.

Since mRNA translation could be affected by the ability of RNA to fold into unique functional structures, we wanted to test whether mRNA transcripts between three tested species (human, *Tetrahymena,* and *Plasmodium*) have a different propensity in RNA folding. This is because RNA secondary structures, especially A-rich sequences, have been shown to coordinate ribosomal frameshifting, impact RNA stability, or slow ribosomal progression to allow protein folding.

[Editors’ note: what follows is the authors’ response to the second round of review.]

Reviewer #2:The revised manuscript addressed the major concerns by performing new experiments (polysome profiling, 2D gels) and by including additional controls. The text and figures are improved, and the story is now more compelling. The remaining issues are fixable and the most significant one is the writing, which still does not offer clear rationales for why some experiments were performed and tends to overinterpret results. In addition, the writing suffers from many typographic errors (some noted below) and some figures can benefit from a better presentation.

We are thankful reviewer #2 for their comments. In the revised manuscript, we have followed majority of the suggestions on improving the text as well as presentation of the data.

1) Figure 4: While the explanation of methodology is improved, I'm not sure what information Figures 4C and 4D add to the story. The text implies that the AT-rich genome could affect structure of mRNAs (less stable). The rationale for this analysis was not clearly stated. It is implied that if evolution has compensated for structure, there would be an effect in the z-scores. But how would this be related to assays of stalling? What is the predicted observation? Since the assay is indirect (structure is not measured) and the purpose is not clearly articulated, I suggest either placing these figures into supplementary or removing them from the manuscript.4) To make Figures 4A and 4B clearer, I recommend directly adding labels for H. sapiens and *P. falciparum* on top of the graphs and improving the figure legends. It's not clear that the first sentence of the legend refers to both A and B. It's not clear what "taken into account" means without context of the main text. Showing full, uncut error bar in Figure 4B and then providing an inset with the y-axis occupancy of 200 may help readers to better grasp this figure since the boxes extending off the plot is misleading. Also, the technical artifact leading to the dip should be acknowledged in the legend. It's not clear why this artifact is not also present in A and therefore whether this artifact is masking the pause in B. To make this more convincing, the authors could create a control dataset where pauses were computationally introduced and then successfully detected (i.e. a positive control). Alternatively, some explanation should be offered in the text as to why this dip does not void this analysis.

In the revised manuscript, we have moved Figure 4C and 4D to supplementary material as a new Figure 4—figure supplement 5. In order to improve presentation of the data we have included new Figures 4A and 4B as well as further explained data presented in Figure 4. and new Figure 4—figure supplement 5. Based on the reviewer suggestion we have also included explanation for the technical artifact in the figure legend.

2) Figure 6: The authors state that previous microarray analysis in Plasmodium showed that Ile starvation does not globally change mRNA levels. Therefore, the rationale to choose this experiment to study NGD in *P. falciparum* is unclear. Why did the authors expect changes on Ile-rich mRNAs given the prior study already showed "similar overall levels for the majority of mRNA transcripts"? Second, is it known that in mammalian cells that aa starvation induces NGD? If so, that should be clearly cited and explained. Alternatively, amino acid starvation could allow uncharged tRNAs to fill the A-site and block Pelota binding, so positive control data in mammalian cells establishing this expectation would add confidence. Third, in yeast, the mRNA stabilizing effects of loss of Dom34 is not that strong and I'm not sure anyone has observed increased mRNA degradation in a Dom34/Pelota overexpression experiment. If such data exists, it should be cited, as it would add confidence. Without offering a stronger rationale for these experiments, the conclusions from this figure are highly speculative and the authors should soften the claims and possibly move it to supplementary data.

The rationale for the Ile-starvation experiment was taken from the previous studies with histidine-induced stalling (using 3-AT, Guydosh and Green, Cell 2014), protein synthesis and subsequent translational arrest during erythroid and platelet maturation (Mills et al., 2016 and Mills et al., 2017), Leu- and Arg-starvation experiments (Darnell et al., Mol Cell 2018) as well as cycloheximide-induced stalling (Simms et al., 2017). We wanted to check whether *Plasmodium* cells react differently during global translational arrest looking more specifically to Ile-rich transcripts as well as levels of NGD factor Pelota and HSP70 chaperone. Based on the reviewer comments and results from Darnell et al., 2018, we have softened our claims and included other explanations for Ile-starvation experiment. We have previously cited Mills et al., 2017, for Pelota overexpression experiments in human cells. This study clearly states that transgenic overexpression of the Pelota in human platelets results in more rapid degradation of most cytoplasmic platelet transcripts (average half-life, 3.7 vs 5.7 hours in control).

3) It is widely accepted that NGD is part of RQC and the authors should clarify that, as previously requested, in the Introduction.

We have now included clarification of RQC and NGD in the Introduction.